# A HER2-Displaying Virus-Like Particle Vaccine Protects from Challenge with Mammary Carcinoma Cells in a Mouse Model

**DOI:** 10.3390/vaccines7020041

**Published:** 2019-05-20

**Authors:** Lisa Nika, Sara Cuadrado-Castano, Guha Asthagiri Arunkumar, Clemens Grünwald-Gruber, Meagan McMahon, Krisztina Koczka, Adolfo García-Sastre, Florian Krammer, Reingard Grabherr

**Affiliations:** 1Department of Biotechnology, University of Natural Resources and Life Sciences, Muthgasse 18, 1190 Vienna, Austria; lisa.nika@boku.ac.at (L.N.); krisztina.koczka@boku.ac.at (K.K.); 2Department of Microbiology, Icahn School of Medicine at Mount Sinai, 1468 Madison Avenue, New York, NY 10029, USA; sara.cuadrado@mssm.edu (S.C.-C.); guha.asthagiri-arunkumar@icahn.mssm.edu (G.A.A.); meagan.mcmahon@mssm.edu (M.M.); adolfo.garcia-sastre@mssm.edu (A.G.-S.); florian.krammer@mssm.edu (F.K.); 3Global Health and Emerging Pathogens Institute, Icahn School of Medicine at Mount Sinai, New York, NY 10029, USA; 4Graduate School of Biomedical Sciences, Icahn School of Medicine at Mount Sinai, New York, NY 10029, USA; 5Department of Chemistry, Division of Biochemistry, University of Natural Resources and Life Sciences, Muthgasse 18, 1190 Vienna, Austria; clemens.gruber@boku.ac.at; 6ACIB—Austrian Centre of Industrial Biotechnology, Muthgasse 11, 1190 Vienna, Austria; 7Department of Medicine, Division of Infectious Diseases, Icahn School of Medicine at Mount Sinai, New York, NY 10029, USA; 8The Tisch Cancer Institute, Icahn School of Medicine at Mount Sinai, New York, NY 10029, USA

**Keywords:** human epidermal growth factor receptor-2 (HER2), baculovirus–insect cell expression vector system (BEVS), virus-like particles (VLPs), cancer vaccine, glycoengineering, AddaVax, Poly (I:C)

## Abstract

Human epidermal growth factor receptor-2 (HER2) is upregulated in 20% to 30% of breast cancers and is a marker of a poor outcome. Due to the development of resistance to passive immunotherapy with Trastuzumab, active anti-HER2 vaccination strategies that could potentially trigger durable tumor-specific immune responses have become an attractive research area. Recently, we have shown that budded virus-like particles (VLPs) produced in *Sf*9 insect cells are an ideal platform for the expression of complex membrane proteins. To assess the efficacy of antigen-displaying VLPs as active cancer vaccines, BALB/c mice were immunized with insect cell glycosylated and mammalian-like glycosylated HER2-displaying VLPs in combination with two different adjuvants and were challenged with HER2-positive tumors. Higher HER2-specific antibody titers and effector functions were induced in mice vaccinated with insect cell glycosylated HER2 VLPs compared to mammalian-like glycosylated counterparts. Moreover, insect cell glycosylated HER2 VLPs elicited a protective effect in mice grafted with HER2-positive mammary carcinoma cells. Interestingly, no protection was observed in mice that were adjuvanted with Poly (I:C). Here, we show that antigen-displaying VLPs produced in *Sf*9 insect cells were able to induce robust and durable immune responses in vivo and have the potential to be utilized as active cancer vaccines.

## 1. Introduction

Human epidermal growth factor receptor-2 (HER2) is overexpressed in a number of different cancer types, including carcinomas of the bladder, ovary, endometrium, pancreas, colon, stomach, gallbladder, esophagus, and breast [1]. HER2 overexpression occurs in 20% to 30% of invasive breast carcinomas and correlates with an aggressive disease phenotype and a poor prognosis [2]. To date, a significant improvement in overall survival in early-stage and metastatic HER2+ breast cancer patients has been achieved due to passive immunization therapy using Trastuzumab, a humanized monoclonal antibody directed against the extracellular domain of HER2, in addition to chemotherapy [3,4,5]. However, despite the benefits displayed by the combined immune–chemotherapy, long-term follow-up data showed that 15% to 24% of the patients experienced tumor recurrence and disease progression, mainly due to the development of resistance to Trastuzumab [6,7]. Hence, there is an urgent need for alternative HER2-targeted immune–therapies that could promote robust and persistent antitumor immune responses without the emergence of resistance. Recent anti-HER2 vaccines focused on triggering a patient’s own immune system to induce tumor-specific antibodies (Abs) have shown promising outcomes [8,9,10]. Among the different vaccine strategies, the use of virus-like particles (VLPs) has shown superiority over monomeric protein antigens or DNA vaccines [11,12,13,14]. VLPs’ advantageous features include higher immunogenicity due to their virus-resembling surface, straightforward antigen delivery, and more effective stimulation of B- and CD4+ T-cell responses due to the high avidity of epitopes on their surface [15,16,17]. Moreover, internalization by antigen-presenting cells has been shown to activate CD8+ T-cells to elicit a specific cytotoxic T-lymphocyte response [18].

Recently, we developed a new VLP platform for complex antigen display based on the baculovirus–insect cell expression vector system (BEVS). The BEVS has been proven to be well suited to the expression of eukaryotic proteins and the production of versatile VLPs [19,20,21]. Although baculovirus-derived VLPs have shown higher immunogenicity in comparison to mammalian cell-based VLPs in vivo, one of the limitations of BEVS lies in the inability to perform the mammalian-like glycosylation required for the production of therapeutic glycoproteins. To address this limitation, we applied SweetBac technology [22] to generate mammalian-like *N*-glycan structures on the antigen. Overall, our VLP technology allowed us to (I) provide HER2 in an authentic conformation and as a complete molecule [23] and to (II) investigate the impact of different glycosylation patterns on the immunogenic effect of HER2-displaying VLPs.

In the present study, we hypothesized that chimeric antigen-displaying VLPs, produced in *Spodoptera frugiperda Sf*9 insect cells, are suitable as cancer vaccines and that differences in glycosylation structures have an impact on antigenicity and antitumor activity in vivo. We generated HER2-displaying VLPs exhibiting insect cell *N*-linked glycosylation composition [23] and novel VLPs, presenting HER2 with a mammalian-like N-linked glycosylation pattern, and investigated immunogenicity and antitumor activity in conjunction with AddaVax (an MF59-like squalene-based oil-in-water nanoemulsion) and Poly (I:C) (polyinosinic–polycytidylic acid, a synthetic analog of double-stranded RNA) adjuvants. Overall, our study sheds a light on future challenges in developing active cancer vaccination strategies, including a determination of the glycosylation pattern on the antigen surface and an assessment of the optimal vaccine adjuvant to generate the most effective antitumor immune response.

## 2. Materials and Methods

### 2.1. Ethics Statement

The animal studies were approved by the Institutional Animal Care and Use Committee at Icahn School of Medicine at Mount Sinai Hospital (IACUC-2014-0234). All research staff were trained in animal care and handling. All efforts were made to minimize animal suffering.

### 2.2. Animals and Cells

Six to eight-week-old female BALB/c mice (albino in-bred mice) were used for the vaccination and tumor challenge studies and were obtained from the Jackson Laboratory (Bar Harbor, ME, USA). The mice were maintained in microisolator cages, kept on a 12-h light and 12-h dark cycle, and had free access to food and water. 

The *Sf*9 insect cells (ATCC CRL-1711) [24] used for the production of recombinant baculoviruses (rBVs) and VLPs were maintained in serum-free HyClone SFM4 Insect cell medium (GE Healthcare, Little Chalfont, UK) supplemented with 0.1% Kolliphor P188 (Sigma-Aldrich, St. Louis, MO, USA) at 27 °C. Suspension cultures were grown in 500-mL shaker flasks at 100 rpm. 

An SK-BR-3 (ATCC HTB-30) human mammary gland cancer cell line expressing human HER2 was grown and propagated in Dulbecco’s modified Eagle’s medium (DMEM; Gibco) containing a penicillin–streptomycin antibiotics mix (100 U/ml penicillin, 100 µg/mL streptomycin; Gibco) and 10% fetal bovine serum (PAN BioTech, P170203), resulting in complete DMEM (cDMEM). 

The TUBO cells (mouse mammary tumor cells) used in our in vivo mammalian cancer syngeneic tumor model were a cloned cell line established in vitro from a lobular carcinoma that arose spontaneously in a BALB-neuT mouse [25]. The TUBO cells were kindly provided by Dr. Federica Carvallo and Dr. Pier-Luigi Lollini (Department of Molecular Biotechnology and Health Science, University of Torino, Italy) and were maintained in cDMEM medium.

### 2.3. Generation of rBV

Here, rBVs encoding full-length human HER2 (GenBank accession No. X03363.1) and the HIV-1 Gag matrix protein (GenBank accession No. K03455.1) were generated using a MultiBac (EMBL, Grenoble, France) expression system, as previously described [23]. In brief, the sequence coding for human HER2 was ligated into the MultiBac acceptor vector pACEBac-1 (EMBL, Grenoble, France), and the sequence coding for the HIV-1 matrix protein was ligated into the pIDC donor vector (EMBL, Grenoble, France). Acceptor–donor fusions were generated by Cre-LoxP recombination and were inserted into the MultiBac genome using competent DH10MultiBac cells according to Fitzgerald et al. [26]. The rBVs were generated by transfecting *Sf*9 cells with the bacmid DNA in conjunction with FuGENE HD transfection reagent (Promega, Madison, WI, USA) according to the manufacturer’s instructions. The viruses were amplified, and viral titers were determined via a tissue culture infectious dose of 50% (TCID_50_): they are given as plaque-forming units per milliliter (PFU/mL). 

### 2.4. Generation and Purification of VLPs

The generation of VLPs, displaying full-length human HER2 with insect cell-specific glycosylation patterns (HER2ic) on the surface, was conducted as previously described [23]. Briefly, rBVs coding for the HIV-1 Gag matrix protein and the full-length human HER2 were used to infect *Sf*9 cells. After harvesting the cell supernatant, VLPs were purified via sucrose gradient ultracentrifugation. Different sucrose gradient fractions were aliquoted and investigated for their HIV-1 Gag and HER2 protein content via western blot analysis. Fractions containing the most target proteins were pooled and investigated by transmission electron microscopy (TEM) and enzyme-linked immunosorbent assay (ELISA) binding studies to confirm the presence of HER2 on the VLP surface in its native conformation.

Moreover, VLPs displaying HER2 with a modified, mammalianized N-linked glycosylation pattern on the surface (HER2ma) were produced. For this, a glycomodule consisting of open reading frames coding for the *Caenorhabditis elegans N*-acetylglucosaminyltransferase II and the bovine β1,4 galactosyltransferase I (controlled by polyhedrin and p10 promoters, respectively) was introduced in the LoxP site of the MultiBac baculovirus genome, as previously described [22]. The resulting backbone was subsequently used as a shuttle vector for the expression of full-length human HER2 integrated into the standard Tn7 site. The generation of VLPs was achieved through coinfection of *Sf*9 cells with rBVs coding for HER2 and glycosyltransferases and rBVs coding for the Gag matrix protein from HIV-1. The parallel expression of target genes and glycosyltransferases modifies the glycan pattern of the proteins expressed in insect cells. VLPs solely comprising of the HIV-1 Gag matrix protein and the host cell surface (Control VLPs) were applied as controls in the following experiments. 

### 2.5. Characterization and Quantification of VLPs

Sucrose-gradient purified VLP samples were mixed with a 4× NuPage LDS sample buffer (Invitrogen, Carlsbad, CA, USA) containing lithium dodecyl sulfate (pH 8.4) and were heated to 70 °C for 10 min. Proteins were separated by sodium dodecyl sulfate-polyacrylamide gel electrophoresis (SDS-PAGE) and were electroblotted onto a polyvinylidene difluoride (PVDF) transfer membrane (GE healthcare Life Sciences, Vienna, Austria). Membranes were probed with primary antihuman HIV-1 Gag rabbit antibody (E63H00201, Enogene, Atlanta, GA, USA), followed by incubation with a secondary antibody (antirabbit IgG alkaline phosphatase-labeled goat antibody; A9919, Sigma-Aldrich, St. Louis, MO, USA). The detection of truncated human HER2 was conducted using Trastuzumab (BioVision, A1046-100) and a secondary antihuman IgG alkaline phosphatase-labeled goat antibody (abcam, ab6858). Full-length human HER2 was detected using anti-HER2 mAb (3B5; Thermo Fisher, Waltham, MA, USA, MA5-13675) and anti-mouse IgG alkaline phosphatase-labeled antibody (Sigma, A3438). The morphology of negatively stained VLPs was determined using TEM (FEI Tecnai G2 200 kV, FEI, Hillsboro, OR, USA) at various magnifications. The VLP concentration was determined by nanoparticle tracking analysis (NTA) using a NanoSight LM-10 (Malvern Instruments Ltd., Worcestershire, UK) [27]. The particle number is given as total particles per mL. The VLPs’ mean diameter and the homogeneity were analyzed by dynamic light scattering (DLS) using a Zetasizer Nano-ZS with software 7.03 (Malvern Instruments, Malvern, UK) at 25 °C. For this, sucrose-gradient purified VLP suspensions were diluted 1:100 in particle-free phosphate buffered saline (PBS) and were measured in five replicates. The size distribution was calculated by the intensity report. The homogeneity of the VLP samples was determined by the polydispersity index (PDI). The total protein concentration of the VLP samples was quantified using Quickstart Bradford Dye Reagent (Bio-Rad, 5000201) using a bovine serum albumin standard curve. To define the HER2 protein content in the VLP samples, a quantification ELISA was performed. Therefore, 96-well plates (Maxisorp; Nunc Thermo Scientific, Waltham, MA, USA) were coated with VLPs and different concentrations of recombinant human HER2 protein overnight at 4 °C. Wells were blocked using PBS containing 0.1% Tween20 (PBS-T) and 0.3% BSA (bovine serum albumin) at room temperature for 1 h. HER2 molecules were stained with 5 µg/mL of Trastuzumab (BioVision, Milpitas, CA, USA, A1046-100) for 1 h at room temperature. Detection was accomplished using an antihuman IgG-horseradish peroxidase (HRP)-conjugated antibody (628420, Invitrogen, Carlsbad, CA, USA), 1:2000 diluted. After washing six times with PBS-T, wells were developed using a 100-µL/well HRP substrate, 3,30,50-tetramethylbenzidine (TMB Stabilized chromogen, Invitrogen, Carlsbad, CA, USA). The reaction was stopped with 100 µL of 2 N H_2_SO_4_, and optical density was measured at 450 nm using an Infinite M1000 plate reader (Tecan Group, Maennedorf, Switzerland). Raw data were analyzed in Microsoft Excel spreadsheets (Version 2003, Microsoft, Redmond, WA, USA). A standard curve for HER2 at known concentrations was established. The relative HER2 protein concentration of the VLP samples was determined using the linear relationship between the absorbance and the HER2 standard. 

### 2.6. Mass Spectrometry Analysis of Complete Glycan Pattern on VLP Surface

Each VLP sample (Control, HER2ic, HER2ma) was transferred into a fresh 1.5-mL screw cap microtube, and cysteine bridges were reduced by the addition of 15 mM of dithiothreitol (Sigma Aldrich, CAS: 3483-12-3) in 100 mM of ammonium bicarbonate buffer and subsequent incubation for 45 min at 56 °C. Samples were then treated with 55 mM of iodoacetamide (Sigma, CAS: 144-48-9) in 100 mM of ammonium bicarbonate (Acros Organics, 393215500) and incubated for 30 min at room temperature in the dark. Proteins were precipitated with acetone (4-fold volume added, 20 min storage at −20 °C) and dried in a SpeedVac (Sorvall). The samples were dissolved in 30 µL of 100 mM ammonium bicarbonate and digested with trypsin (Promega, V511A 20 µg) at 37 °C overnight (enzyme to substrate ratio of 1:50). To destroy any residual protease activity, the samples were incubated at 95 °C for 10 min. One-hundred micro-litres of Peptide N-Glycosidase A (PNGase) A buffer (citric acid monohydrate 244, Merck Darmstadt, Germany, sodium dihydrogen phosphate, J.T. Baker, 303; pH 5) and 0.75 µL (=11.25 mU) of PNGase A (Proglycan) per sample were then added, and the samples were incubated at 37 °C overnight. The released glycans were dried in a SpeedVac (Sorvall) and reconstituted in 90 µL of 100-mM ammonium bicarbonate buffer. Ten micro-litres of 10% sodium borohydride solution (Sigma Aldrich, 452882) were added and incubated overnight at room temperature. The digested glycans were purified using HyperSep Hypercarb solid phase extraction (SPE) 10-mg cartridges (Thermo Scientific, 60302606). The elution of bound glycans was performed using 500-µL SPE of 60% acetonitrile. The samples were dried in a SpeedVac (Sorvall), redissolved in 20 µL of high-quality water, and subjected to liquid chromatography-electrospray ionization mass spectrometry (LC-ESI-MS). The digested samples were loaded on a porous graphitic carbon column (100 × 0.32 mm, Thermo Scientific) using 80 mM of ammonium formiate buffer (Ammonia solution, VWR, CAS: 1336-21-6; Formic acid, Acros Organics, CAS: 64-18-6) as aqueous solvent and acetonitrile (VWR, CAS: 75-05-8) as solvent B. A gradient from 2% to 42% solvent B (98%–58% solvent A) was developed over 20 min at a flow rate of 8 μL/min using an Ultimate 3000 capillary flow LC (Dionex, Thermo Fisher, Waltham, MA, USA). With this steep gradient, all glycans eluted within a time window of a few minutes, which facilitated depiction of the whole glycan profile. Detection was performed using quadrupole time of flight (QTOF) MS (Bruker maXis 4G) equipped with a standard ESI source in positive ion data dependent acquisition (DDA) mode (=switching to tandem mass spectrometry mode for eluting peaks). MS-scans were recorded in a range from 150 to 2200 Da. Data interpretation and quantification was performed with DataAnalysis 4.0 (Bruker, Billerica, MA, USA) and QuantAnalysis 2.0 (Bruker). The extracted ion chromatograms of the first four isotopic mass peaks of every detected glycoform were integrated. Glycoforms were quantified by comparing peak areas. As shown by Grünwald-Gruber et al. [28], signal responses of detected glycan variants are in a similar range, and thus comparing peak areas is valid.

### 2.7. Immunization and In Vivo Tumor Challenge 

For the immunization studies, 6–8-week-old female BALB/c mice (*n* = 5 mice/group) were vaccinated intramuscularly (IM) with 5 µg total protein of Control VLPs, HER2ic VLPs, and HER2ma VLPs in a prime-boost regimen at day 1 and day 21. For each immunization, vaccines were adjuvant with 5 µg Poly (I:C) (InvivoGen, tlrl-pic) or 25 µL AddaVax (InvivoGen, vac-adx-10) per dose or were left nonadjuvant, resulting in nine different vaccination groups. Four weeks after the boost vaccination, 2.5 × 10^5^ TUBO cells were intradermally (ID) implanted into the flank of the right posterior leg of each mouse and under light anesthesia. Starting from the day when tumors were first palpable, tumor growth was monitored every 48 h for the extent of the study. Tumor measurement was determined using a digital caliper, and total volume was calculated using the formula tumor volume *(V) = L × W × W/2*, where *L*, or tumor length, is the larger diameter, and *W*, or tumor width, is the smallest. Animals that bore a tumor larger than 1000 mm^3^ or displayed an open necrotic lesion were humanely euthanized by inhalation of 2% CO_2_. Growth curves were plotted for each vaccination group. Blood was collected the day before tumor implantation (4 weeks after the booster vaccination) and 3 weeks after the tumor challenge by submandibular bleeding. Blood samples were centrifuged for 10 min at 2000× *g* in a microcentrifuge, and the serum was stored at −20 °C and was used for ELISA, an antibody-dependent cellular cytotoxicity (ADCC) reporter assay, and IgG subtype determination. 

### 2.8. Histological Analysis of Tumor Biopsies

Representatives of each experimental group were humanely euthanized once the tumors reached the experimental end point of 1000 mm^3^. Long-term survivors were euthanized at day 70 post-tumor implantation. Excision biopsies were preserved by formalin fixation and paraffin embedding for immunohistochemistry analysis (IHC). A histological evaluation was carried out at the Comparative Pathology Laboratory, Center of Comparative Medicine and Surgery at Icahn School of Medicine at Mount Sinai, New York (US). IHC staining of CD8+ lymphocytes was performed on 5-µm-thick tumor sections. A polyclonal, mouse-specific anti-CD8 antibody (Abcam, Cambridge, UK; ab203035) was used following the manufacturer’s instructions. The final counterstaining was performed using hematoxylin.

### 2.9. Serum Antibody Responses

To measure the serum antibody responses of vaccinated mice, ELISAs were conducted. Ninety-six-well flat-bottomed immunoplates (Thermo Fisher) were coated with 50 µL of recombinant human HER2 protein (expressed in High Five insect cells) at a concentration of 2 µg/mL in phosphate buffered saline (PBS, Gibco) at 4 °C overnight. The following day, 220 µL of blocking solution, PBS supplemented with 0.1% Tween-20 (PBS-T, Fisher Scientific), and 0.3% low-fat milk powder (American-Bio, Canton, MA, USA) were added to all wells of the microtiter plate and were incubated for 1 h at room temperature. Serum Abs were serially diluted 1:2, starting with an initial dilution of 1:50 in the first well, and were incubated for 2 h at room temperature. After washing three times with PBS-T, 50 µL per well of antimouse IgG (whole molecule)-HRP conjugated antibody (produced in rabbit, Sigma, #A9044) diluted 1:3000 in blocking buffer was added and incubated for 1 h at room temperature. The plates were washed four times with PBS-T and were developed by adding 100 µL/well of SigmaFast o-phenylenediamine dihydrochloride (OPD, Sigma). The reaction was stopped using 50 µL/well of 3 M hydrochloric acid (Thermo Fischer), and plates were read at 490 nm using a microtiter plate reader (Bio-Tek, Winooski, VT, USA). Raw data were analyzed in Microsoft Excel (Version 2003) and GraphPad Prism 7 and were normalized to blank.

### 2.10. Determination of Induced Anti-HER2 IgG Subtypes

To investigate the IgG subtypes of the induced anti-HER2 antibodies in mouse serum, an ELISA was performed as described previously (see Section 2.9). In brief, recombinant human HER2 (2 µg/mL in PBS) was coated on 96-well MaxiSorp plates (Thermo Fisher) overnight at 4 °C. Afterwards, the plates were blocked using PBS-T supplemented with 0.3% milk powder (AmericanBio) for 1 h at room temperature. Serum samples were two-fold serially diluted and incubated for 2 h at room temperature. After washing three times with PBS-T, anti-mIgG1-HRP (abcam, ab97240), anti-mIgG2a-HRP (abcam, ab97245), and anti-mIgG2b-HRP (abcam, ab97250) antibodies (1 mg/mL) were diluted 1:3000 in blocking buffer and were added for 1 h at room temperature. Plates were washed four times and were developed using OPD substrate (Sigma). The enzymatic reaction was stopped by adding 100 µL of 3-M hydrochloric acid (Thermo Fischer), and optical density was measured at 490 nm using a plate reader (BioTek). Raw data were analyzed in Microsoft Excel (Version 2003) and GraphPad Prism 7. Results were normalized to blank and are plotted as IgG1:IgG2a:IgG2b ratios of total induced anti-HER2 Abs. 

### 2.11. ADCC Reporter Assay

An ADCC reporter bioassay kit (BioGlo Luciferase Assay System, Promega, G720B, G7198) was applied to assess whether any of the anti-HER2 serum Abs elicited ADCC activity. The assay was carried out according to the manufacturer’s instructions. In brief, SK-BR-3 (ATCC HTB-30) cells (5 × 10^4^/well) were seeded onto white 96-well cell culture plates (Corning Costar, 3917) and were incubated for 48 h at 37 °C, 5% CO^2^. Twofold serial dilutions of each mouse serum Ab sample were added to the cells, starting from an overall dilution of 1:50. Seventy-five-thousand effector cells, expressing the murine Fc_γ_ receptor IV (mFc_γ_RIV), were subsequently added to every 96-well containing the SK-BR-3 cells and the antibody dilutions. Cells were incubated at 37 °C for 6 h. Luciferase substrate was added, and luminescence was measured after 5 min incubation using a Synergy Hybrid Reader (BioTek). Raw data were processed using a Microsoft Excel spreadsheet (Version 2003) and GraphPad Prism 7 and are shown as the area under the curve (AUC). Significant differences between groups were analyzed using a standard *t*-test.

To assess whether there was a correlation between ADCC activity in vitro and tumor growth in vivo, the ADCC activity, as the AUC, was plotted over the tumor volume on the day the serum samples were taken. Correlations are indicated by the Pearson correlation coefficient (*r*^2^) and significance values. 

### 2.12. Assessment of T-Cell Activation

#### 2.12.1. Processing of Spleens

Mice were immunized as previously described (see Section 2.7). Eighteen days after the boost vaccination, spleens were harvested from the mice and were pushed through a 70-µM strainer (Foxx Life Sciences, Salem, NH, USA). The splenocytes were then centrifuged at 1600 rpm for 5 min (HERAEUS Megafuge16, Thermo Scientific, Waltham, MA, USA), and red blood cells were lysed in ammonium-chloride-potassium (ACK) lysis buffer comprised of 0.15 M NH_4_Cl (Sigma A-4514, MW = 53.49), 10 mM KHCO_3_ (Sigma P-9144), and Na_2_EDTA (Sigma ED2SS), pH 7.2. Splenocytes were then centrifuged at 1600 rpm for 5 min and were resuspended in complete α-medium (MEM α-modification; Sigma M4526, 10% FCS; Sigma F-2442, 100 U penicillin/100 µg streptomycin mix; Sigma P-0781, 4 mM l-glutamine; Sigma G-7513, 50 µM 2-Mercaptoethanol; Gibco BRL 31350-010) at a cell concentration of 2.5 × 10^6^ cells/mL.

#### 2.12.2. Enzyme-Linked Immunospot (ELISpot) Assay

An interferon (IFN)_γ_ ELISpot assay was performed according to the manufacturer’s instructions (MabTech, 3321-2A). Briefly, ELISpot plates (EMD Millipore, MAIPS4510) were coated with 15 µg/mL antimouse IFN_γ_ capture antibody. The following day, splenocytes (2.5 × 10^6^ cells/mL) were added to the plate and were incubated at 37 °C for 48 h in the presence or absence of 1 µg/mL full-length recombinant human HER2 protein. After 48 h, the plates were washed and incubated with the biotinylated mouse anti-IFN_γ_ monoclonal detection antibody (R4-6A2, 1mg/mL) diluted in PBS containing 0.5% FBS (PBS-0.5% FBS) for 2 h at room temperature. Following the secondary incubation, plates were stained with HRP-conjugated streptavidin (1 mg/mL in PBS-0.5% FBS) for 1 h at room temperature. Visualization of IFN_γ_-producing cells occurred by using a BioRad AP substrate kit (BioRad, 1706432) per the manufacturer’s instructions, and spots were counted in an ImmunoSpot Analyzer 5.0.3 Pro DC (Cellular Technology, LLC). Raw data were processed using Microsoft Excel (2003) and GraphPad Prism 7. Significant differences between groups were analyzed using a standard *t*-test.

#### 2.12.3. Intracellular Cytokine Staining (ICS)

One-hundred micro-litres of cells (2.5 × 10^6^ cells/mL) per sample were stimulated for 5 h at 37 °C using 1 µg/mL full-length recombinant human HER2 protein in the presence of Brefeldin A Solution (BioLegend, San Diego, CA, 420601). Unstimulated samples were included as controls. After stimulation, splenocytes were washed in fluorescence-activated cell sorting (FACS) staining buffer (PBS supplemented with 0.1% bovine serum albumin) and stained for anti-CD4 (Clone RM4.4, 116013) and anti-CD8 (Clone 53-6.7; BioLegend, 140414) T-cell cell surface markers. Antibodies were incubated with cells for 30 min at 4 °C and were subsequently washed in FACS buffer and fixed using fixation buffer (BioLegend, 420801) according to the manufacturer’s instructions. Following fixation, splenocytes were washed in permeabilization buffer (BioLegend, 421002) and stained for IFN_γ_ (Clone XMG1.2, BioLegend, 505805) expression for 30 min at 4 °C. Afterwards, cells were sequentially washed with permeabilization buffer, resuspended in 200 µL FACS buffer, and immediately analyzed on a modified LSR II flow cytometer (BD Biosciences, Franklin Lakes, NJ, USA). Data were analyzed using FlowJo v10 and were presented using GraphPad Prism 7.0.

### 2.13. Statistical Analysis

Data analysis was performed using Excel (Version 2003) and GraphPad Prism 7.0 software (GraphPad Software Inc., San Diego, CA, USA). The unpaired nonparametric *t*-test was used to assess differences between two groups. A *p*-value <0.05 was considered statistically significant (* *p* < 0.05, ** *p* < 0.01, *** *p* < 0.001, **** *p* < 0.0001). A two-way Anova–Tukey multiple comparisons test was used for the analysis of tumor growth over time. A survival curves analysis was performed using the long-rank (Mantel–Cox) test (** *p* = 0.0061). Linear correlations between ADCC activity and tumor volumes are given by the Pearson correlation coefficient (*r*^2^).

## 3. Results

### 3.1. Generation and Characterization of HER2ic and HER2ma VLP Vaccines

Chimeric budded VLPs were generated in *Sf*9 insect cells (Figure 1A) infected with rBVs expressing the HIV-1 Gag matrix protein and the full-length human HER2 protein following a procedure, as previously described [23]. By applying that system, HER2 displayed on the VLP surface exhibited *N*-linked glycopatterns specific for insect cells (HER2ic VLPs). For the production of VLPs that expressed HER2 with a mammalian-like *N*-linked glycopattern, *Sf*9 cells were coinfected with rBVs coding for *Caenorhabditis elegans N*-acetylglucosaminyltransferase II, bovine β1,4-galactosyltransferase I, and full-length human HER2 and rBVs coding for the Gag matrix protein from HIV-1 (Figure 1A). Using that technique, VLPs displaying recombinant HER2 with mammalianized complex-type *N*-glycans were generated (HER2ma VLPs). VLPs solely comprised of the Gag matrix protein were applied as a negative control in the following experiments (Control VLPs).

The morphology, particle integrity, and expression of control, HER2ic, and HER2ma VLPs were examined by TEM and western blot analysis (Figure 1B–E). Sucrose-density purified chimeric VLPs appeared as spherical with a size approximately 100–200 nm in diameter (Figure 1B–D). Western blot analysis using antihuman HER2 mAb (3B5), Trastuzumab, and anti-HIV-1 Gag rabbit antibody, as shown in Figure 1E, confirmed the expression and incorporation of full-length human HER2 and HIV-1 Gag into VLPs. Full-length HER2 was detected as a protein of 138 kDa, concurring with the calculated molecular protein mass. An additional specific band was detected at 65 kDa, matching the HER2 cleavage product (HER2_trunc) [29]. Control VLPs did not show any HER2 protein expression. HER2 with mammalianized glycosylation appeared heavier than the insect cell glycosylated counterparts, indicating the presence of additional glycans. HIV-1 Gag was detected at 55 kDa, which coincided with the theoretical molecular protein mass, and was present in all three chimeric VLP preparations.

The presence of terminal galactose residues of proteins displayed on the VLP surface was confirmed by analyzing *N*-glycans originating from glycosidase (PNGase A) treatment with LC-ESI-MS. In Figure 2A–C, the MS spectra of the released glycans are shown. The major glycoforms were oligomannosidic (Man4-Man9+1Hex) and paucimannosidic structures (MM, MMF). Moreover, other glycoforms such as MGn (F) and GnGnF were detected and are shown as relative proportions in Table 1. In the glycoengineered strain (HER2ma VLPs), a clear shift in the complete glycosylation pattern could be seen (Figure 2C). A major part of the insect cell-type MMF was modified to the complex-type mammalian structure AAF (G2F). The peak heights roughly reflected the molar ratios of the glycoforms. In Table 1, the quantitation of the different glycoforms is shown (quantified by integration of the extracted ion chromatogram (EIC) of the first four isotopic peaks). Proglycan nomenclature was used (http://www.proglycan.com/protein-glycosylation-analysis/nomenclature). Since a whole glycopattern analysis was done, it was difficult to assess whether the glycans were derived from the VLPs, the contaminating baculoviruses, or any other protein impurity in the vaccine preparations. Still, for detecting alterations in glycan patterns toward mammalian-like structures, all *N*-glycosylated proteins displayed the desired modifications, suggesting that the VLPs also displayed proteins with mammalianized alterations.

In Table 2, a detailed quantification of the VLP vaccine preparations is shown. The particle concentrations were determined by NTA and ranged from 6–8 × 10^11^ particles/mL for particles with a diameter between 100 and 200 nm (62%–76% of total particles). The quality of the preparations was determined by dynamic light scattering (DLS), and the mean diameter and homogeneity were calculated. All particles had a mean diameter size of 140 ± 10 nm, and the calculated PDI indicated a moderate polydisperse nanoparticle distribution (PDI 0.1–0.4; for DLS diagrams, see Appendix A). Baculovirus titers of the vaccine preparations were determined by TCID_50_ assay adapted for insect cells, and they ranged from 4.9 × 10^8^–1.1 × 10^9^ PFU/mL. Total protein yields were defined by densitometry. Moreover, HER2 protein yields were assessed by quantitative ELISA using recombinant HER2 protein as the standard, and they were approximately twice as high for HER2ic VLPs (62.62 µg/mL) compared to HER2ma VLPs (29.10 µg/mL).

### 3.2. Prophylactic Immunization and Tumor Challenge

Due to the potent capacity of VLPs to stimulate humoral and cellular immune responses [15], we hypothesized that priming an individual’s immune system with VLPs carrying tumor-specific antigens prior to tumor implantation would be efficient in preventing or ameliorating the progression of antigen-positive tumors. To test our hypothesis, we immunized mice with Control VLPs and HER2 VLPs displaying either insect cell-specific glycans (HER2ic) or mammalianized glycopatterns (HER2ma). Moreover, we investigated the potential of two different adjuvants, Poly (I:C) and AddaVax. The tumor challenge was performed by grafting immunized mice with HER2+ mammary carcinoma cells (TUBO cells [25]), and the emergence and growth of syngeneic lesions was monitored over time (Figure 3A). For comparative purposes, an experimental control group of non-immunized, age-matching mice was also subjected to implantation of TUBO cells at the time of the tumor challenge (Figure 3A). Once the mice reached the experimental end point, they were humanly euthanized, and two representatives of each group were submitted for excision biopsy (Figure 3E; Appendix A).

Mice vaccinated with Control VLPs showed a tumor development and progression similar to the non-immunized control group, with an overall survival of 20–22 days post-implantation (Figure 3B–D). A histological analysis of tumor biopsies described the development of solid mammary adenocarcinomas, which were highly vascularized, with significant mitotic activity (16–18 per 400× field) and a variable necrosis index (≤25%–50% per 400× field) at the time of excision. An immunohistochemistry analysis revealed rare infiltration of lymphocytes (0–2 per 400× field), with CD8+ cells mainly retained in the peripheral stroma (Figure 3C,E). Prophylactic immunization with insect cell glycosylated HER2ic VLPs was effective in delaying tumor emergence and shaping tumor development in both nonadjuvant and AddaVax-adjuvant cohorts when compared to the rest of the experimental groups. Statistically significant differences in the control of tumor growth and long-term survival were detected by day 20 postimplantation and were sustained over time (Figure 3C,D). By day 70 postimplantation, 20% of the mice immunized with HER2ic VLPs and included in nonadjuvant and AddaVax-adjuvant groups showed stable disease with positive progression to complete remission. A histological analysis of tumor biopsies from long-term survivors revealed the presence of neoplasms with lower mitotic activity (≤10 per 400× fields) and necrosis indexes (≤25% per 400× field) and a higher presence of lymphocytes and myeloid cells infiltrating the tumor mass (Figure 3C,E). The presence of CD8+ cells was higher in AddaVax HER2ic VLPs than in the nonadjuvant counterparts, with individual and small CD8+ cell aggregates often observed infiltrating neoplastic cords, especially those adjacent to the stroma. Notably, no protective effect was achieved in the Poly (I:C) HER2ic VLP group. Additionally, no protection was detected in all groups immunized with mammalian-like glycosylated HER2ma VLPs, regardless of the adjuvant used. A histological analysis from these groups did not differ from the control groups, but slightly higher variations in the necrosis index (25%–50% per 400× field) were observed in the Poly (I:C)-adjuvant tumor samples.

### 3.3. Anti-HER2 Antibody Response in Sera and IgG Subtype Determination

To evaluate the antibody response of vaccinated mice, pre- and post-tumor challenge serum samples were tested in an ELISA against recombinant human HER2 protein. As shown in Figure 4A, mice within the HER2ic and HER2ma vaccine groups induced high Ab titers when compared to control groups. Notably, HER2ma VLP-vaccinated mice showed lower Ab induction compared to their insect cell glycosylated counterparts in all three adjuvant groups (nonadjuvant, Poly (I:C), and AddaVax). Overall, the AddaVax HER2ic VLP-immunized group induced the highest anti-HER2 Ab titers. No major changes were observed three weeks after tumor implantation. Furthermore, binding of the serum antibodies to recombinant mouse HER2 was confirmed in an ELISA (Appendix A), proving the cross-reactivity of antibodies induced by VLP vaccinations. When testing the induced serum antibodies against the carrier protein antigen (Control VLPs), comparably low titers were obtained with all immunizations, indicating that the immune response was specifically directed toward the HER2 target epitope as opposed to the carrier protein (Appendix A). 

To analyze the IgG subtypes elicited by HER2ic and HER2ma VLP vaccination, an ELISA against recombinant human HER2 using secondary Abs for IgG1, IgG2a, and IgG2b subclass detection was conducted. As shown in Figure 4B, the IgG1 subtype comprised the lowest quantity of total IgG in all HER2 VLP-vaccinated groups. The highest titers were obtained with IgG2a, followed by IgG2b (for raw data, see Appendix A). These subclasses of IgGs are known to be the most proinflammatory in mice and have been described as having higher biological activity than IgG1 subtypes [30]. In general, no major changes in the IgG subtype composition between the different vaccination groups pre- and post-immunization were observed.

### 3.4. In Vitro Effector Function of Induced Anti-HER2 Antibodies

To investigate whether the induced anti-HER2 Abs elicited effector functions that could possibly drive antitumor effects, an ADCC assay was carried out. Possible interactions between mFc_γ_RIV and the Fc regions of antibodies bound to HER2-expressing tumor cells can be detected and quantified using that assay. As shown in Figure 5A, most Abs induced by HER2ic VLP vaccination elicited significantly higher ADCC activity in vitro compared to Control VLPs in all three vaccination groups (nonadjuvant, Poly (I:C), AddaVax) in both pre- and post-tumor implantation conditions. Moreover, HER2ic VLP-immunized mice showed increased ADCC activity when compared to HER2ma VLP-vaccinated mice. The highest levels were obtained in the nonadjuvant and AddaVax-adjuvant HER2ic VLP-immunized groups, which were also the groups that elicited tumor protection. The control groups did not show any activity in vitro. 

To assess whether the in vivo antitumor effects were driven by effector functions of induced anti-HER2 Abs, correlations between tumor volumes in vivo and ADCC activity in vitro were plotted for each experimental group. As shown in Figure 5B, a significant correlation (*** *p* < 0.001) was observed in the nonadjuvant vaccination group, exhibiting an *r*^2^ of 0.6247. No significant correlation was found in either the AddaVax group (*r*^2^ = 0.1739, *p* = 0.1380) or the Poly (I:C) group (*r*^2^ = 0.0048, *p* = 0.8294). Nevertheless, a clear trend could be observed within the AddaVax cohort. When plotting all groups together, a significant correlation was obtained, with an *r*^2^ of 0.1885 and ** *p* < 0.01, suggesting that the antitumor effect in vivo was partly driven by effector functions of anti-HER2 Abs induced after VLP immunization. 

### 3.5. IFN_γ_ Response of HER2-Stimulated T-Cells

As durable protective responses toward HER2+ tumors require T-cell functions [31], CD4+ and CD8+ T-cell responses were examined after HER2ic VLP vaccination. Mice were immunized with 5 µg total protein per dose of Control VLPs and HER2ic VLPs (nonadjuvant, AddaVax) in a prime-boost regimen, and T-cell responses were assessed 18 days later using an ELISpot assay and FACS analysis. In Figure 6A, IFN_γ_-secreting splenocytes after HER2 stimulation are shown. Both nonadjuvant and AddaVax-adjuvant HER2ic VLP vaccination groups showed increased IFN_γ_ responses compared to control groups. As can be seen in Figure 6B, VLP immunization induced HER2-specific CD4+ and CD8+ T-cell responses, as measured by intracellular IFN_γ_ cytokine production. While there were no overt differences in CD4+IFNy+ T-cell responses, significantly higher CD8+IFN_γ_ responses (* *p* < 0.05) were induced in the AddaVax HER2 VLP-vaccinated groups in comparison to Control VLP-vaccinated groups. Furthermore, nonadjuvant HER2 VLP immunization increased CD8+IFN_γ_ responses compared to Control VLPs. These results suggest that vaccination with HER2ic VLPs significantly increased the population of HER2-specific CD4+ and CD8+ T-cells (*n* = 5).

## 4. Discussion

Passive immunotherapy with anti-HER2 monoclonal antibodies (such as Trastuzumab) alongside chemotherapy is the standard of care for HER2-positive breast cancer patients. However, resistance to these treatments and a restricted duration of action due to the short half-life and elimination rate of the applied antibodies remain an issue to be overcome. Taking into account these limitations, active anti-HER2 vaccination strategies have become an attractive therapeutic approach. Such strategies could potentially trigger tumor-specific immune responses, induce long-lasting immunity against tumor antigens, and reduce severe side effects [8,9].

In the present study, we tested antigen-displaying enveloped VLPs expressed in *Sf*9 insect cells as vaccine candidates. It has been shown that the composition of glycans attached to therapeutic proteins potentially influences the efficacy and immunogenicity of the drug [32]. However, the impact of the glycosylation in the case of HER2 studies remains poorly explored. We generated chimeric HIV-1 Gag VLPs expressing full-length HER2 with insect cell-specific glycosylation on the surface [23] as well as a variant displaying HER2 with mammalianized glycan structures by deploying SweetBac technology [22]. To investigate the glycan composition of our VLP preparations, we applied LC-ESI-MS. The major glycoforms detected in the HER2ic VLPs were oligomannosidic and paucimannosidic structures. In the glycoengineered strain (HER2ma VLPs), a major part of the insect cell-type MMF was modified to the more complex-type mammalian structure AAF (G2F). 

For our in vivo tumor challenge, we used the TUBO cell line [25], a HER2+ cloned cell line established in vitro from a spontaneous mammary adenocarcinoma developed by a BALB/c-neuT mouse, which is transgenic for the rat *HER2-neu* oncogene [33]. TUBO-based in vivo syngeneic tumor models have been extensively used in the field of anti-HER2 DNA vaccine development, allowing HER2+ tumors to arise in the receptor animal without rejection and facilitating structural and functional anti-HER2 in vivo studies without off-target effects [25,33]. Although rat HER2 displays 95% of the amino acidic sequence homology of mouse HER2 (85% in the case of the human protein), and cross-reactivity of induced anti-HER2 antibodies was confirmed, this slight difference in amino acid composition between proteins constituted a limitation in our model and should be taken into consideration in order to evaluate the positive and future potential of our prophylactic strategy, especially when it pursues antigen-specific responses.

In order to achieve an adaptive, long-term, protecting antitumor response in vivo, our prophylactic strategy must be able to induce at first an effective innate immune response that could lead to an efficient tumor-specific antigen presentation. In that matter, the relatively high residual baculovirus contamination present in our VLP preparations could stimulate innate immune responses via the toll-like receptor 9 (TLR9), hence contributing to building up the antigen-specific antitumor response [34].

When we compared the antitumor activity of HER2ic and HER2ma VLP immunizations in our in vivo model, higher protection was induced by insect cell glycosylated HER2ic VLPs compared to mammalian-like glycosylated HER2ma VLPs. HER2ic VLP vaccination was able to delay tumor emergence, control tumor growth, and extend survival significantly, with 20% of nonadjuvant and AddaVax-treated mice displaying stable disease. Strikingly, no protection was achieved with mammalian-like glycosylated VLP vaccinations. It has been previously described that expression of HER2 in *Drosophila* S2 insect cells has the potential advantage of improving antigen uptake by antigen presenting cells (APCs) due to the addition of insect glycans. Specifically, the paucimannose insect glycan structure has been reported to induce improved uptake by dendritic cells and other APCs due to the expression of membrane-bound mannose receptors in these cells [14,35,36,37]. In addition, in our study, the presence of high levels of paucimannose insect cell glycans could be beneficial for antigen uptake by APCs and therefore enhance the antitumor effect elicited by HER2ic VLPs. In general, we clearly showed that there was a significant difference in the antitumor response induced by the VLP vaccines that displayed insect cell or mammalian-like glycosylated antigens. The Control VLPs did not show any antitumor effects.

We have also revealed that the use of Poly (I:C) as an adjuvant did not contribute to achieving an efficient antitumor response in our VLP-based prophylactic strategy. The synthetic double-stranded RNA analog is known to exert a strong inflammatory response by interacting with toll-like receptor 3 (TLR3). TLR3 is expressed in the membranes of B-cells, macrophages, and dendritic cells, all three of them with very well-established APC capacity. Although triggering TLR3 signaling induces the activation of multiple proinflammatory pathways (as NFκB and IRF3 signals) [38], it is likely that in our system the baculovirus component in combination with Poly (I:C) strongly drove the innate response toward a favorable recognition of viral components rather than toward the tumor antigen, due to the synergetic effect of the activation of TLR9 and TLR3. Thus, the antigen-specific antitumor response mediated by HER2+ VLPs could be diminished when administered with Poly (I:C). On the other hand, AddaVax is a squalene-based oil-in-water emulsion that elicits both cellular (Th1) and humoral (Th2) immune responses. AddaVax is believed to act through recruitment and activation of APCs and through stimulation of cytokine production by macrophages and granulocytes [39]. It has been previously shown that cytokines such as IL-12 are required for the establishment of long-term protection and a tumor-specific memory against mammary adenocarcinomas, which are mediated by CD8+ lymphocytes [33]. As we have previously noted, HER2ic-AddaVax-immunized tumor biopsies have shown an increased presence of tumor-infiltrating CD8+ lymphocytes that could contribute to the long-term response observed in this group. 

When we investigated the underlying tumor-specific immune responses, we observed that HER2ic VLP immunizations induced higher anti-HER2 Ab titers than the mammalianized counterparts. In any case, the induced antibodies were biased toward the more active IgG2a and IgG2b subtypes. These murine IgG2 subtypes have been described as having higher biological activity than IgG1 or IgG3 antibodies [30]. Accordingly, the induced anti-HER2 Abs mediated ADCC effector functions. The highest activity was detected in HER2ic VLP-immunized mice that were given AddaVax or were left nonadjuvant. In addition, both of these vaccination groups showed the highest tumor protection, and hence showed significant correlations between the ADCC activity in vitro and reduced tumor volumes in vivo. In the Poly (I:C) cohort, no correlation was observed. This strongly suggests that the antitumor effect was partly driven by effector functions of the induced HER2-specific Abs. 

Since it is known that IgG2a and IgG2b Ab subtypes mostly drive ADCC effector functions [30], we anticipated higher IgG2a and IgG2b levels than IgG1 levels in the vaccination groups that showed high ADCC activity. As we did not observe any obvious changes in the subtype ratios among the different vaccination groups, we suggest that the induced anti-HER2 antibodies might display different glycosylation patterns on their Fc regions that lead to differences in binding to Fc_γ_ receptors on natural killer (NK) cells [40], therefore leading to differences in the intensity of the ADCC activity.

Moreover, T-cell activity was clearly increased in HER2 VLP-immunized mice when compared to Control VLP-immunized mice, indicating that cell-mediated immune responses were also involved in the tumor protective effects. However, more research needs to be addressed to confirm that.

We are also aware of the fact that using the HIV-1 Gag matrix protein in a human vaccine might be problematic, since it could trigger nonspecific immune responses and cause side effects. Yet, there exist a variety of alternative matrix proteins, such as M1 from the influenza virus [41], that form VLPs and that could be used when given to patients.

## 5. Conclusions

In summary, our results demonstrated that *Sf*9 cell-derived chimeric VLPs, comprised of the HIV-1 Gag matrix protein and the cancer antigen HER2, had potent effects of inducing durable humoral and cellular immune responses in vivo. We further demonstrated the importance of assessing the best adjuvant and showed that cell type-specific glycan structures had an impact on the efficacy of a potential vaccine. Both of these parameters must be evaluated carefully and optimized in order to generate an effective and durable immune response. Thus, we consider these results as a direction for future cancer immunotherapy.

## Figures and Tables

**Figure 1 vaccines-07-00041-f001:**
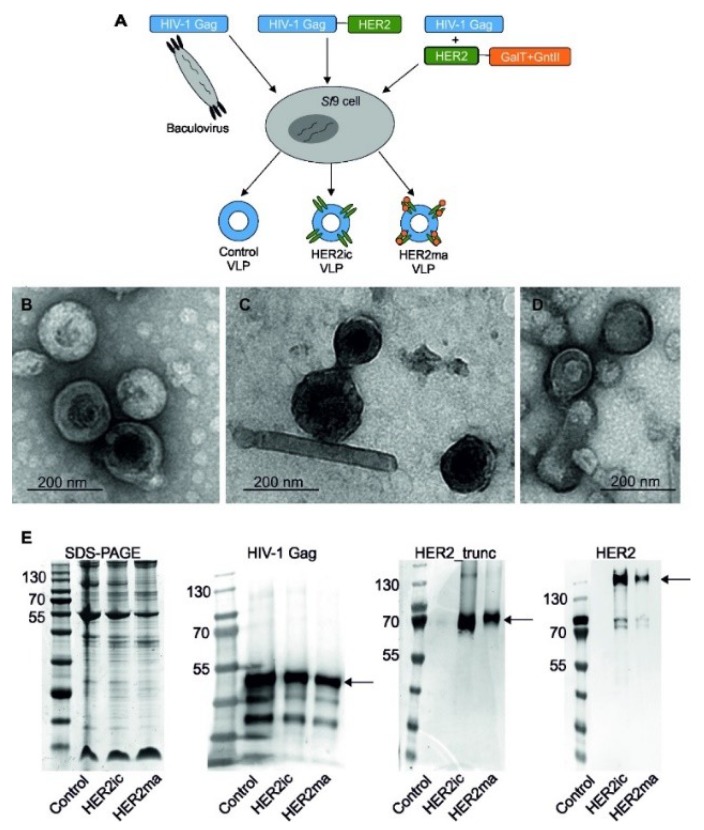
Infection schematic and characterization of Control, HER2ic, and HER2ma virus-like particle (VLP) vaccines. In (**A**), the schematic of recombinant baculoviruses (rBVs) used for the generation of VLPs is shown. In (**B**–**D**), sucrose-density purified VLPs were negatively stained and investigated using a transmission electron microscope (FEI Tecnai G2 200 kV, FEI, Hillsboro, OR, USA). All types of VLPs ((**B**) the control, (**C**) HER2ic, (**D**) HER2ma) appeared spherical. In (**E**), the sodium dodecyl sulfate-polyacrylamide gel electrophoresis (SDS-PAGE) and western blot analysis results of the VLP preparations are shown. Trastuzumab, a monoclonal anti-HER2 antibody (3B5), and a monoclonal rabbit antihuman HIV-1 Gag antibody were used to probe truncated HER2, full-length human HER2 protein, and the HIV-1 Gag matrix protein, respectively.

**Figure 2 vaccines-07-00041-f002:**
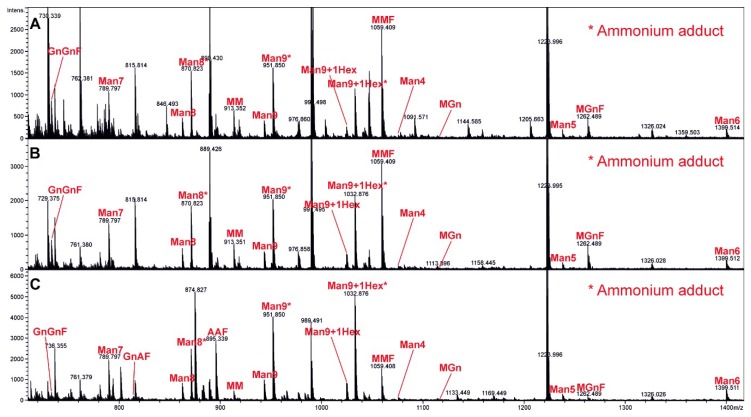
MS spectra of the released and reduced *N*-glycans. (**A**) Control, (**B**) HER2ic, and (**C**) HER2ma VLP vaccines were processed, treated with glycosidase, and investigated for their complete glycan pattern by liquid chromatography-electrospray ionization mass spectrometry (LC-ESI-MS) analysis. The highest glycan peaks are annotated.

**Figure 3 vaccines-07-00041-f003:**
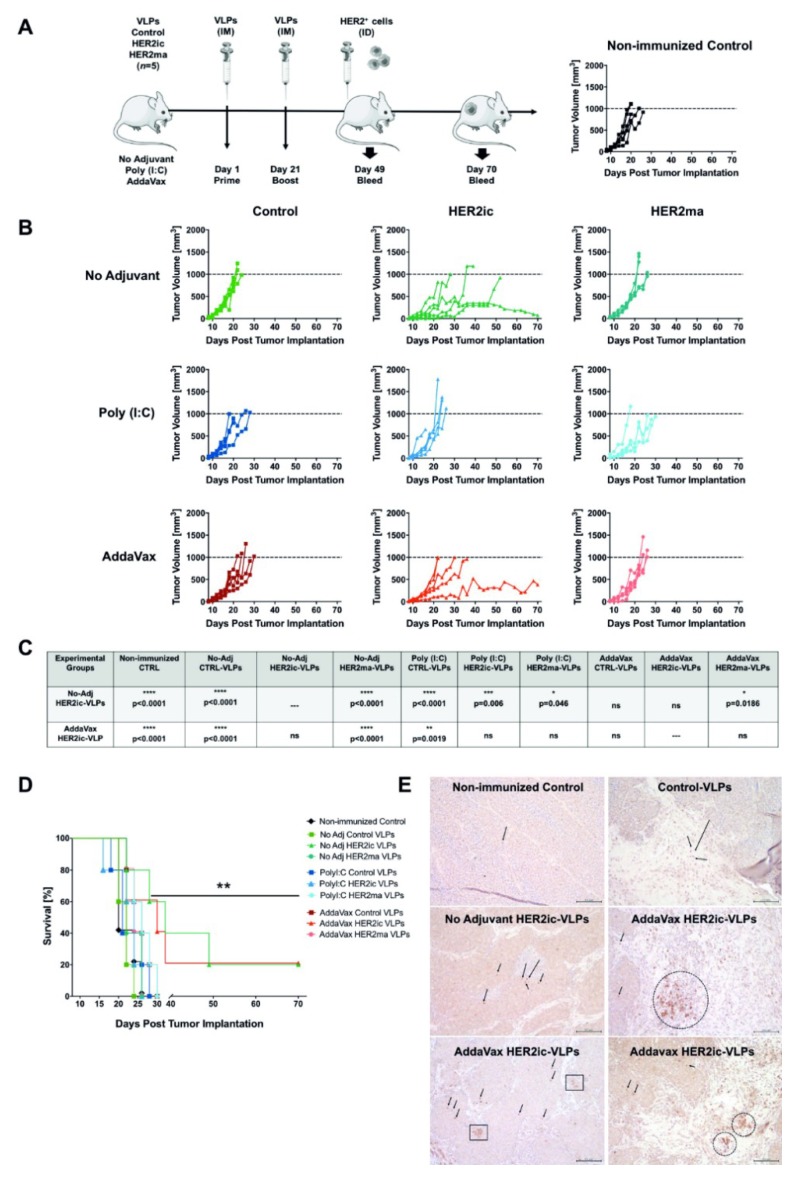
In vivo antitumor efficacy of prophylactic administration of HER2 VLPs. In (**A**), the study design is shown. Six to eight-week-old female BALB/c mice were vaccinated intramuscularly (IM) in a prime-boost regimen. Each experimental cohort (no adjuvant (I), polyinosinic–polycytidylic acid (Poly (I:C)) (II), and AddaVax (III)) was further divided into three subgroups (*n* = 5): Control VLPs, HER2ic VLPs, and HER2ma VLPs. Four weeks after boosting, HER2+ mammary carcinoma cells were implanted intradermally (ID) into the flank of the right posterior leg of immunized mice. Blood samples were taken submandibularly just before tumor implantation and three weeks after. (**B**) Tumor progression (I): individual tumor growth curves, with each dot representing tumor volume per mouse at the indicated time point. The dotted line on the tumor volume axis represents the experimental endpoint (1000 mm^3^). (**C**) Tumor progression (II): statistical analysis of tumor progression based in tumor volume measurements by day 20 post implantation; two-way Anova–Tukey multiple comparison results summary for the HER2ic and HER2ma VLP groups are shown. (**D**) Long-term survival: overall survival of each experimental group (** *p* = 0.0061). (**E**) Histopathology: immunohistochemistry revealed the presence and distribution of CD8+ T-lymphocytes in 5-µm-thick tumor sections. Black arrows note individual CD8+ T-cells. Squares highlight areas of CD8+ T-cell aggregates. Rounded lines define areas with an abundance of CD8+ T-cells and myeloid cells. Magnification 200×; scale bar 100 µm.

**Figure 4 vaccines-07-00041-f004:**
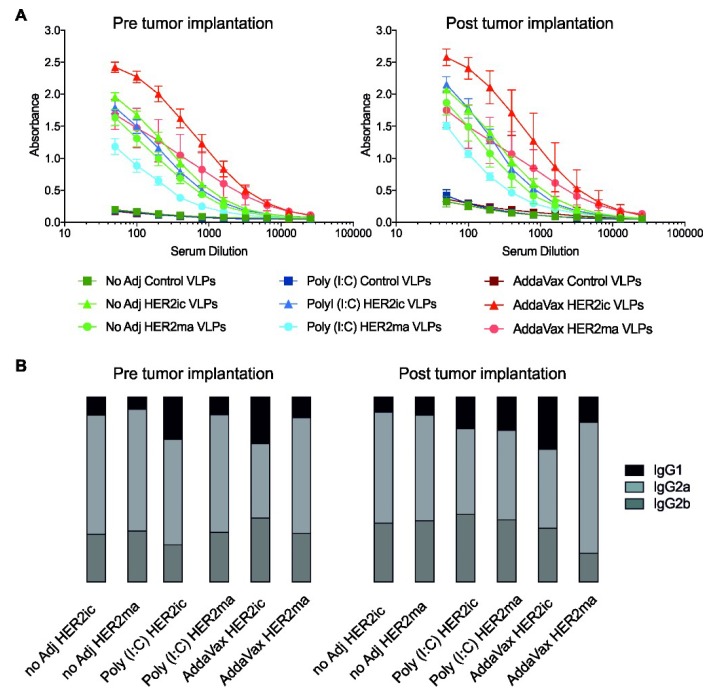
Antibody responses upon immunization with Control, HER2ic, and HER2ma VLPs. Mice were immunized with Control, HER2ic, or HER2ma VLPs in a prime-boost regimen, in a nonadjuvant, or in combination with Poly (I:C) or AddaVax. (**A**) Serum antibodies of vaccinated mice were investigated in an ELISA against recombinant HER2 protein. Results are shown as mean ± SEM (standard error of mean) of each vaccination group (*n* = 5). In (**B**), IgG1:IgG2a:IgG2b subtype ratios of induced anti-HER2 antibodies elicited by the indicated vaccinations are shown.

**Figure 5 vaccines-07-00041-f005:**
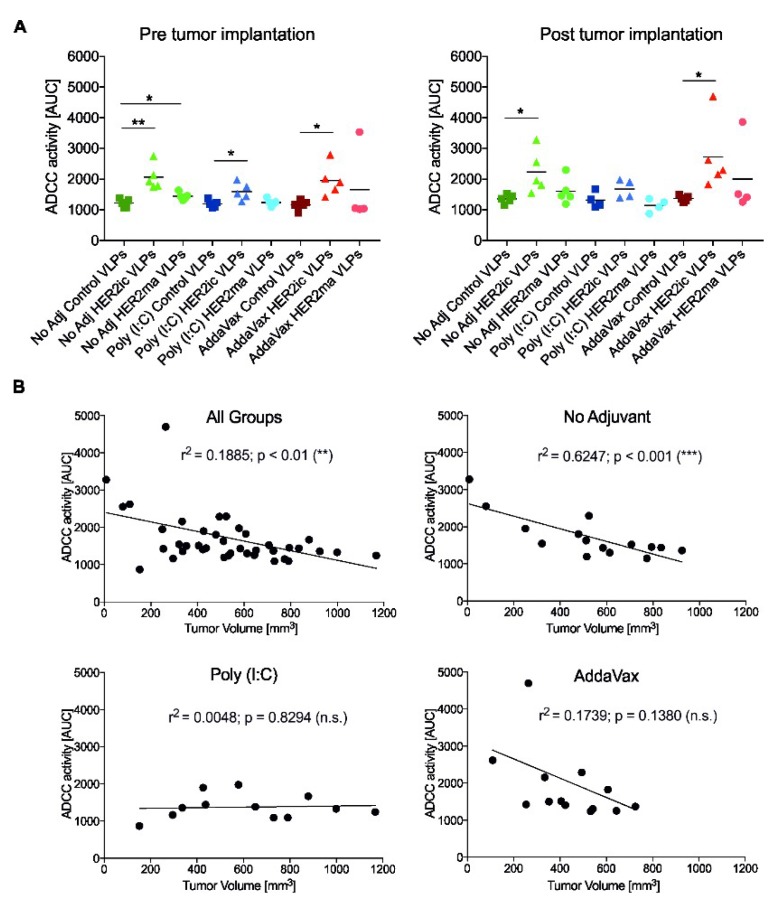
Effector functions of induced anti-HER2 antibodies. In (**A**), the antibody-dependent cellular cytotoxicity (ADCC) activities of induced anti-HER2 antibodies pre- and post-tumor implantation are shown. The results are given as the area under the curve (AUC). Every symbol represents the AUC of a single mouse (*n* = 5). Differences between Control VLP and HER2ic/HER2ma VLP vaccinations were assessed using an unpaired nonparametric *t*-test. In (**B**), the correlations between tumor volumes in vivo and ADCC activity in vitro are shown. Tumor volumes are indicated from the day of serum sample collection (21 days post-implantation). Every dot represents the correlation of a single mouse. The Pearson correlation coefficient (*r*^2^) and significance values are indicated on the graph (* *p* < 0.05, ** *p* < 0.01, *** *p* < 0.001).

**Figure 6 vaccines-07-00041-f006:**
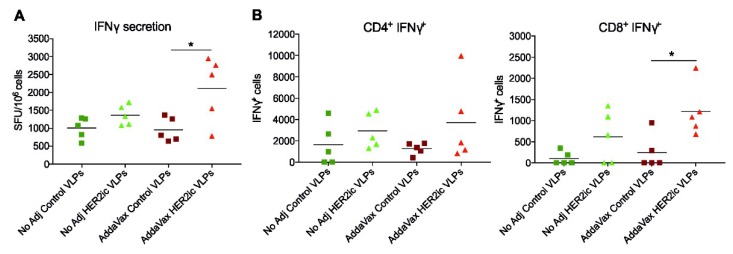
T-cell induction and cytokine response after VLP immunization. Mice were vaccinated with 5 µg of Control VLPs and HER2ic VLPs in a prime-boost manner. Eighteen days after the boost vaccination, splenocytes were isolated and investigated for T-cell induction by (**A**) interferon (IFN)_γ_ secretion at a single cell level and (**B**) intracellular IFN_γ_ production. In (**A**), the results were obtained through enzyme-linked immunospot (ELISpot) assay (MabTech 3321-2A) and are shown as spot-forming units (SFUs) per 10^6^ cells. In (**B**), the results were obtained through intracellular cytokine staining (ICS) and are displayed as CD4+ and CD8+ IFN_γ_+-producing cells. Every dot represents the signal of a single mouse (*n* = 5). Differences between Control VLP and HER2ic VLP vaccinations were assessed using an unpaired nonparametric *t*-test (* *p* < 0.05).

**Table 1 vaccines-07-00041-t001:** LC-ESI-MS analysis of glycans originating from glycosidase treatment ^1^.

Glycoform	Percent of Total (%)		
	Control	HER2ic	HER2ma
Man4	0.48	0.5	0.42
Man5	3.53	2.99	2.95
Man6	6.43	6.48	7.82
Man7	11.88	11.14	9.89
Man8	14.97	14.32	11.93
Man9	17.83	17.63	20.85
Man9 + Hex	14.17	18.77	26.17
MM	3.38	2.64	1.52
GnM	0.57	0.46	0.62
MMF	15.66	15.11	4.31
GnMF	3.11	2.99	0.48
GnGnF	6.73	5.38	0.95
AGnF	-	-	0.68
AAF	1.26	1.4	11.41

^1^ The results are shown as the proportion of glycoforms in %. Proglycan nomenclature was used (http://www.proglycan.com/protein-glycosylation-analysis/nomenclature).

**Table 2 vaccines-07-00041-t002:** Quantification of VLP vaccine preparations.

VLP Vaccine	Concentration ^1^	100–200 nm ^2^	Size ^3^	Baculovirus Titer ^4^	Total Protein ^5^	HER2 Protein ^6^	Polydispersity Index ^7^
	(Particles/mL)	(% of Total)	(d.nm)	(PFU/mL)	(µg/mL)	(µg/mL)	PDI (0.1–0.4)
Control	7.97 × 10^11^	76.59	140.00	6.77 × 10^8^	1780	/	0.146
HER2ic	6.07 × 10^11^	62.08	146.30	1.14 × 10^9^	840	62.62	0.155
HER2ma	6.36 × 10^11^	70.57	132.10	4.90 × 10^8^	620	29.10	0.154

^1^ The particle concentration was measured by nanoparticle tracking analysis (NTA) using a NanoSight LM-10 (Malvern Instruments Ltd., Worcestershire, UK).^2^ Particles with a diameter between 100 and 200 nm were selected and are shown as % of total particles.^3^ The size of the particles was measured by dynamic light scattering (DLS) (Zetasizer, Malvern). The results are shown as diameters in nm (particle size distribution by number).^4^ The baculovirus titers of the VLP vaccine samples were determined by a tissue culture infectious dose of 50% (TCID50) assay and were converted into plaque-forming units (PFUs) per mL.^5^ The total protein concentration was quantified by Quickstart Bradford Dye Reagent (Bio-Rad, 5000201) with a bovine serum albumin standard curve.^6^ Total HER2 protein was determined by quantitative ELISA.^7^ The polydispersity index (PDI) was determined by dynamic light scattering (Zetasizer, Malvern). A PDI between 0.1 and 0.4 refers to a moderate polydisperse suspension.

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
