# Peer review of "A HER2-Displaying Virus-Like Particle Vaccine Protects from Challenge with Mammary Carcinoma Cells in a Mouse Model"

_vaccines, 2019, doi:10.3390/vaccines7020041_

Round 1

Reviewer 1 Report

This manuscript describes the use of enveloped VLP made from the HIV GAG-MA structural expressed in insect cells for the presentation of the HER2 protein, a well known lung and breast cancer target.

The approach is interesting and shows some promising results. However, there are some major issues concerning the vaccine preparation and composition that need to be clarified. The major objective of the manuscript was to assess the gain in immunogenicity by adding the HER2 at the surface of a VLP. As a secondary objective, the authors wish to assess the influence of the glycosylation pattern of the VLPs on the immunogenicity. Finally, the final aim is to determine that capacity of such a VLP presenting HER2 to induce protection in a mouse cancer model.

Comments:

The adequate analysis of the composition of the different samples used for immunization is critical and they should be comparable to draw appropriate conclusions. 

a) Figure 1: A SDS-PAGE for each of the three different VLPs preparation should be showed with the identification of each of the expected proteins. This is important to assess the level of contaminants in the vaccine preparation. It is an easy way to compare the content of the three samples.

b) Figure 1: The quality of the western blot is not acceptable. A full gel should be showed for the western blots directed to HER2 and to the HIV Gag to assess for the degradation products and the overall quality of the sample that have been immunized.

c) The cleavage product of HER2 is of different molecular weight between the HER2ic and the HER2ma. Is it real or is it a gel effect because one of the two samples was overloaded?

d) Figure 2: Several glycans (like Man 4 to 6) described at table 1 are not identified at Figure 2. Either the author should delete them from the table because they are not present in sufficient amount to be discussed or their position at Fog 2 should be highlighted like the others.

            Also, there is no way to assess if the glycans issued from this digestion are derived from the VLP, the contaminant baculovirus or any other contaminating proteins in the vaccine preparation. Therefore, it is very difficult to draw any conclusion from this assessment.

e) Table 2: The curve of the DLS should be presented. It is anticipated that at least two peaks will be observed, on for the VLPs (about 100nm) and one for the contaminating baculovirus (about 260nm). The authors should demonstrate the homogeneity of the samples. The curves can be placed in Suppl. data.

f) Table 2: This table reveal that HER2ic contains at least 2 times more contaminant baculovirus than the HER2ma. As mentioned in the discussion, the baculovirus can influence the immunogenicity of the samples. To be comparable, the amount of contaminating baculovirus between the two samples should be standardized and made equivalent between the samples. Otherwise, it is impossible to assess the effect of the glycosylation pattern on the immunogenicity.

g) In addition, this table reveal that there is at least 2 times less HER2 protein associated with the VLP of HER2ma than HER2ic. It is likely that a 2-fold difference in the content of HER2 between Her2ma and HER2ic can have a major impact on the immune response that will be triggered. Therefore, a group of mice immunized with at least 2 times more HER2ma should have been included in the study to balance the amount of HER2 injected in the animals.

f) Figure 3: The quality of the image is too low and it is difficult to see the results presented. 

g) To highlight the benefice of the presentation of HER2 onto by the VLP, a control group with purified HER2 alone should be included. As it is presented here, it is not possible to judge if the VLP are improving the immune response directed to HER2.

h) What is the proportion of the HER2 that is located at the surface of the VLP with the HER2ic and HER2ma constructs and how much of HER is free I in the vaccine preparation? This should be investigated and answered to conclude on the potential of the VLP in the improvement of the immune response to HER2.

h) The adjuvant Addavax is a oil in water adjuvant (MF-59 like adjuvant). Since the VLPs are enveloped with the cell membrane and since the enveloped is necessary to present the HER2 protein at their surface, the impact of the adjuvant on the structure of the VLP should be investigated. Does it disrupt the membrane of the VLPs? Does it release the HER2 from the VLP? Does it chance the antigenic presentation of the HER2 antigen?

i) The authors chose to use only 5 mice per group for the immunization and the assessment of the anti-cancer efficacy of the vaccine. The experiment should at least include 10 mice per group and be repeated twice to provide a reasonable level of confidence.

j) There is 85% homology between the human and the mice HER2. Are the differences located in the surface exposed region of the protein? Are the antibody directed to the human HER2 capable to bind to the mouse HER2? This is critical to understand the mechanism of action of the vaccine candidates. This should be demonstrated. 

k) What is the contribution of the CTL response directed toward HER2 in the anti-cancer activity recorded? Could you repeat this experiment in mice where the CD8+ cells are depleted before implantation of the cancer cells?

Author Response

Response to Reviewer 1 Comments

Point 1: Figure 1: A SDS-PAGE for each of the three different VLPs preparation should be showed with the identification of each of the expected proteins. This is important to assess the level of contaminants in the vaccine preparation. It is an easy way to compare the content of the three samples.

Response 1: We have included a SDS-PAGE gel (Fig.1E), showing HIV-1 Gag and its degradation products and contaminants. Since merely the HIV-1 Gag in VLPs can be clearly seen in Coomassie staining and HER2 only in Western Blot analysis, we have additionally focused in more detail on the quantification of HER2 protein on the surface of VLPs in this work (indicated in Table 2).

Point 2: Figure 1: The quality of the western blot is not acceptable. A full gel should be showed for the western blots directed to HER2 and to the HIV Gag to assess for the degradation products and the overall quality of the sample that have been immunized.   

Response 2: We replaced Figure 1 E by a SDS-PAGE gel showing HIV-1 Gag (see above) and three full Western Blots showing HIV-1 Gag, HER2 and HER2_trunc.The Figure legend (line 366) was changed to: “(E) SDS-PAGE and Western blot analysis results of the VLP preparations are shown. Trastuzumab, a monoclonal anti-HER2 antibody (3B5) and monoclonal rabbit anti-human HIV-1 Gag antibody were used to probe truncated HER2, full-length human HER2 protein and HIV-1 Gag matrix protein, respectively.

Line 151 was rephrased: “…Aldrich, St. Louis, Missouri, USA). Detection of truncated human HER2 was conducted using…”

A sentence was added in line 153: “Full-length HER2 was detected using anti-HER2 mAb (3B5; Thermo Fisher, MA5-13675) and anti-mouse IgG alkaline phosphatase-labeled antibody (Sigma, A3438).”

Line 351 was rephrased: ”…analysis using anti-human HER2 mAb (3B5), Trastuzumab and anti HIV-1 Gag rabbit antibody, as…”

Point 3: The cleavage product of HER2 is of different molecular weight between the HER2ic and the HER2ma. Is it real or is it a gel effect because one of the two samples was overloaded?

Response 3: Results indicate that glycosylation is responsible for the difference in size since glyco-analysis showed a defined difference between mammalianized and insect cell glycosylated preparations (indicated in Table 1 and Figure 2), and calculated molecular weight is in accordance with the experimental results. Moreover we have never before observed a shift of HER2 in molecular weight, due to overloading; here same amounts of VLPs have been loaded onto the gel.

Point 4: Figure 2: Several glycans (like Man 4 to 6) described at table 1 are not identified at Figure 2. Either the author should delete them from the table because they are not present in sufficient amount to be discussed or their position at Fig 2 should be highlighted like the others.

Also, there is no way to assess if the glycans issued from this digestion are derived from the VLP, the contaminant baculovirus or any other contaminating proteins in the vaccine preparation. Therefore, it is very difficult to draw any conclusion from this assessment

Response 4: Figure 2 was changed accordingly. All glycans present in Table 1 were annotated in Figure 2.

We agree that it is difficult to assess whether glycans are derived from the VLP, baculovirus or any other contaminating protein in the vaccine preparations, however for detecting alterations in glycan patterns towards mammalian like structures, all N-glycosylated proteins display such modifications, whether derived from VLP, baculovirus or any other contaminant. We can therefore assume that VLPs also display proteins with mammalianized modifications. Moreover, the sample subjected to MS analysis was exactly the same as was injected into mice. All identified glycans were present in the preparations used for vaccination.  

Point 5: Table 2: The curve of the DLS should be presented. It is anticipated that at least two peaks will be observed, on for the VLPs (about 100nm) and one for the contaminating baculovirus (about 260nm). The authors should demonstrate the homogeneity of the samples. The curves can be placed in Suppl. data

Response 5: We have included the curves of the DLS as supplementary material (Supplementary Figure 1); however, just a single peak is visible, due to the log scale on the x-axis, which is automatically generated by the Zetasizer softwear. In our study we wanted to demonstrate homogeneity of the sample, no cell debris or bigger aggregates are visible. The curves indicate a high quality biological sample, PDI is included in table 2 and is below 0,2 confirming homogeneity.

Line 394 was rephrased: “…VLPs were homogeneously distributed (PDI < 0,2; DLS diagrams see Supplementary Figure 1).

Point 6: Table 2: This table reveal that HER2ic contains at least 2 times more contaminant baculovirus than the HER2ma. As mentioned in the discussion, the baculovirus can influence the immunogenicity of the samples. To be comparable, the amount of contaminating baculovirus between the two samples should be standardized and made equivalent between the samples. Otherwise, it is impossible to assess the effect of the glycosylation pattern on the immunogenicity.

Response 6: We agree, that significant differences in BV concentration might influence the immune response, however based on our experience we strongly believe it’s highly unlikely that a 2-fold BV concentration is biologically significant. Studies in the past have shown (Margine et al., 2012, DOI:10.1371/journal.pone.0051559) that there is a difference whether BV are present or completely absent, but small differences in BV concentrations in vaccine samples had so far minor to no influence.

Moreover, vaccine preparations were standardized based on total protein concentration, see Response 7.

Point 7: In addition, this table reveal that there is at least 2 times less HER2 protein associated with the VLP of HER2ma than HER2ic. It is likely that a 2-fold difference in the content of HER2 between Her2ma and HER2ic can have a major impact on the immune response that will be triggered. Therefore, a group of mice immunized with at least 2 times more HER2ma should have been included in the study to balance the amount of HER2 injected in the animals

Response 7: VLPs were standardized based on total protein content. Total protein content of HER2ma was lower than HER2ic (indicated in Table 2); therefore higher amounts were vaccinated, suggesting roughly the same content of HER2 in the vaccine preparations. According to ELISA of serum samples, we detected high anti-HER2 antibody titres in both, HER2ic and HER2ma vaccinated animals. Moreover, immune responses do not follow a linear pattern and small differences in HER2 are not expected to cause a significant difference in the immune response.

Point 8: Figure 3: The quality of the image is too low and it is difficult to see the results presented

Response 8: Figure 3 was replaced accordingly.

Point 9: To highlight the benefice of the presentation of HER2 onto by the VLP, a control group with purified HER2 alone should be included. As it is presented here, it is not possible to judge if the VLP are improving the immune response directed to HER2

Response 9: We agree that it would be beneficial to show the improvement of the immune response when using VLP-displayed HER2 compared to purified HER2. However, in our study we only compared different VLP samples in conjunction with different adjuvant substances, and can’t judge whether our samples are more effective as compared to recombinant HER2. Yet, there are studies (Palladini et al., 2018, doi:10.1080/2162402X.2017.1408749) that have demonstrated that the immune response to VLPs was higher as compared to HER2 DNA vaccines.

Point 10: What is the proportion of the HER2 that is located at the surface of the VLP with the HER2ic and HER2ma constructs and how much of HER is free I in the vaccine preparation? This should be investigated and answered to conclude on the potential of the VLP in the improvement of the immune response to HER2

Response 10: We agree that knowing the proportions of VLP incorporated HER2 and free HER2 would be beneficial to conclude on the potential of the VLP in the improvement of the HER2 specific immune response. Although, after ultracentrifugation we do not expect a significant quantity of free HER2 to be present in the samples. To clarify, detailed characterization of the VLP samples in context with their immune effects will be subject to future studies.

Point 11: The adjuvant Addavax is a oil in water adjuvant (MF-59 like adjuvant). Since the VLPs are enveloped with the cell membrane and since the enveloped is necessary to present the HER2 protein at their surface, the impact of the adjuvant on the structure of the VLP should be investigated. Does it disrupt the membrane of the VLPs? Does it release the HER2 from the VLP? Does it chance the antigenic presentation of the HER2 antigen?

Response 11: Such detailed studies would provide a valuable contribution, however are beyond the scope of the presented work. Oil in water suspension is similar to lipid membranes which surround the VLPs and should not be destructive, yet even if their structure is destroyed, we would expect that still intact HER2 would be present in the samples. Still, the question is highly interesting and follow up studies will be done in order to investigate the impact of AddaVax on the structure of the VLP.

Point 12: The authors chose to use only 5 mice per group for the immunization and the assessment of the anti-cancer efficacy of the vaccine. The experiment should at least include 10 mice per group and be repeated twice to provide a reasonable level of confidence

Response 12: The mouse experiments were exploratory, we agree that more mice would be more representative, however our sample was enough to be hypothesis generating in nature, follow-up studies with successful candidates will use a higher number of mice.

Point 13: There is 85% homology between the human and the mice HER2. Are the differences located in the surface exposed region of the protein? Are the antibody directed to the human HER2 capable to bind to the mouse HER2? This is critical to understand the mechanism of action of the vaccine candidates. This should be demonstrated

Response 13: There is 85% homology between the extracellular domains and 88% between the full-length human and mouse HER2.

For better understanding and clarification we have investigated binding of serum antibodies to recombinant mouse HER2 and included ELISA data as supplementary material (Supplementary Figure 3). Mouse serum against human HER2 showed binding to mouse HER2, indicating that human anti HER2 antibodies recognize recombinant mouse HER2 and are cross-reactive. In contrast to monoclonal antibodies, immunized sera contain a large variety against the antigen.

We have included following sentence in the results section (line 476): “Furthermore, binding of serum antibodies to recombinant mouse HER2 was confirmed in an ELISA (Supplementary Figure 3), proofing the cross-reactivity of antibodies induced by VLP vaccination.

The following was added in line 577: “…and cross-reactivity of induced anti-HER2 antibodies was confirmed,…

Point 14: What is the contribution of the CTL response directed toward HER2 in the anti-cancer activity recorded? Could you repeat this experiment in mice where the CD8+ cells are depleted before implantation of the cancer cells?

Response 14: We have addressed this issue in the Discussion (see line 614):

“It has been previously shown that cytokines like IL-12 are required for the establishment of a long-term protection and tumor-specific memory against mammary adenocarcinomas which is mediated by CD8+ lymphocytes [33]. As we have previously noted, HER2ic-AddaVax immunized tumor biopsies have shown an increased presence of tumor infiltrating CD8+ lymphocytes that could contribute to the long-term response observed in this group.”

Experiments with CD8+ depleted mice would be very interesting, however cannot be accomplished within the scope of this work, but will be subject of further studies.

Reviewer 2 Report

This is a very interesting pice of work, presenting a strategy for vaccine development for cancer treatment. It is based on the proven epitope from the antibody drug approaches.The epitope of interests, HER2, was grafted to antigen-displaying VLPs. These recombinant proteins are used as active cancer vaccines to generate effective anti-tumor immune response. New generation molecular adjuvants were used to further enhance the immunogenicity.

Below are few questions/points that could help to further improve the story:

1.     HER2ic or HER2ma VLPs with vaccine adjuvant, Addavax, were shown to elicit higher antibody response. However, it did not help the anti-tumor response.  In addition, HER2ic with poly I:C adjuvants showed no protective effect in mice grafted with HER2-positive mammary carcinoma cells. Thus, please add the data of the immune response or titers on the target antigen (HER2) and carrier protein antigen, and the whole antigen , respectively. This may help to explain the higher antibody titers but with lower anti-tumor activity. The possible scenario could be the immune response was directed more toward the carrier protein, as opposed to the desired HER2 epitope.

2.     No protection was achieved with mammalian-like glycosylated VLP vaccinations. Are mammalian-like glycosylation in antigens necessary? Any report on the criticality on the glycans on the HER2 epitopes? This point can be further elaborated in discussion for the experiment design and for the results.

3.    It would be helpful if the observation period of antibody response should be further prolonged.

 Minor comments-

1.     Figure 1E. MW markers should be added to the figure.

2.  Figure 3 is of low quality. The information presented is unclear, especially the letters for data labels. Please make appropriate changes to improve the figure.

3. The raw data of IgG1/2 could be shown as Supplementary Materials. Please consider.

Author Response

Response to Reviewer 2 Comments

Point 1: HER2ic or HER2ma VLPs with vaccine adjuvant, Addavax, were shown to elicit higher antibody response. However, it did not help the anti-tumor response. In addition, HER2ic with poly I:C adjuvants showed no protective effect in mice grafted with HER2-positive mammary carcinoma cells. Thus, please add the data of the immune response or titers on the target antigen (HER2) and carrier protein antigen, and the whole antigen , respectively. This may help to explain the higher antibody titers but with lower anti-tumor activity. The possible scenario could be the immune response was directed more toward the carrier protein, as opposed to the desired HER2 epitope

Response 1: We compared VLPs without HER2 on the surface, VLPs with HER2ic and HER2ma on the surface, which means that on all VLPs we have the insect cell membrane surrounding the HIV-Gag protein; the difference between the samples is the presence and glycosylation pattern of the HER2. HER2 specific antibody response was detected whenever HER2 was present on the surface. VLPs without HER2 induced no HER2 response. The fact that HER2 response did not automatically correspond with anti tumor effects can be explained by the different ADCC activities detected in the different samples (indicated on Figure 5). Antibody titers against the target antigen (HER2) are indicated on Figure 4.

We agree that investigating the immune response against the carrier protein antigen would be a valuable contribution to our study. Therefore, serum samples were also tested against control VLPs (no HER2). Results revealed comparable low antibody titers against the insect cell membrane in all immunized mice. Specific HER2 response was only detected when HER2 was present in the vaccine preparation. For better understanding and clarification, ELISA data showing binding of serum antibodies to control VLPs were added as supplementary material (Supplementary Figure 4).

A sentence was added in the Results section (line 478): “When testing induced serum antibodies against the carrier protein antigen (Control VLPs), comparable low titers were obtained with all immunizations, indicating that the immune response is specifically directed toward the HER2 target epitope as opposed to the carrier protein (Supplementary Figure 4).“

Point 2: No protection was achieved with mammalian-like glycosylated VLP vaccinations. Are mammalian-like glycosylation in antigens necessary? Any report on the criticality on the glycans on the HER2 epitopes? This point can be further elaborated in discussion for the experiment design and for the results

Response 2: There was a scientific report published in 2017 on the effect of glycosylation on the binding of drugs to cell surface glycoproteins (HER2) and on sensitivity of cancer cells to chemotherapeutic agents (Peiris et al., 2017, DOI: 10.1038/srep43006). But overall, the impact of glycosylation on the HER2 epitope in the context of therapeutics remains poorly understood.

For clarification we rephrased the following paragraph starting at line 559 in the discussion section: “In the present study we tested antigen-displaying enveloped VLPs expressed in Sf9 insect cells as vaccine candidates. It has been shown that the composition of glycans attached to therapeutic proteins potentially influences the efficacy and immunogenicity of the drug [32]. However, the impact of the glycosylation in case of HER2 studies remains poorly explored. We generated chimeric HIV-1 Gag VLPs expressing full-length HER2 with insect cell specific glycosylation on the surface [23] as well as a variant displaying HER2 with mammalianized glycan structures by deploying the SweetBac technology [22]

Point 3: It would be helpful if the observation period of antibody response should be further prolonged.

Response 3: We agree, that prolonging the observation of the antibody response would be helpful, however in our experiments we determined antibody titers before the tumor challenge and three weeks after the tumor challenge, which we consider sufficient for the proof of our concept. In future studies however, we will consider a prolongation.

Point 4: Figure 1E. MW markers should be added to the figure

Response 4: Full Western blots and molecular weights were added. We replaced Figure 1 E by a SDS-PAGE gel and three full Western Blots showing HIV-1 Gag, HER2 and HER2_trunc. The Figure legend (line 366) was changed to: “(E) SDS-PAGE and Western blot analysis results of the VLP preparations are shown. Trastuzumab, a monoclonal anti-HER2 antibody (3B5) and monoclonal rabbit anti-human HIV-1 Gag antibody were used to probe truncated HER2, full-length human HER2 protein and HIV-1 Gag matrix protein, respectively.

Line 151 was rephrased: “…Aldrich, St. Louis, Missouri, USA). Detection of truncated human HER2 was conducted using…”

A sentence was added in line 153: “Full-length HER2 was detected using anti-HER2 mAb (3B5; Thermo Fisher, MA5-13675) and anti-mouse IgG alkaline phosphatase-labeled antibody (Sigma, A3438).”

Line 351 was rephrased: ”…analysis using anti-human HER2 mAb (3B5), Trastuzumab and anti HIV-1 Gag rabbit antibody, as…”

Point 5: Figure 3 is of low quality. The information presented is unclear, especially the letters for data labels. Please make appropriate changes to improve the figure

Response 5: Appropriate changes were made and Figure 3 was replaced accordingly.

Point 6: The raw data of IgG1/2 could be shown as Supplementary Materials. Please consider

Response 6: Raw data of IgG subtype analysis was added as supplementary material (Supplementary Figure 5).

In line 486 the following was added: “…(raw data see Supplementary Figure 5)…”

Reviewer 3 Report

In the study by Nika et al the authors have used VLPs to display Her2 tumor antigen in an attempt to generate an immune response. They used HIV-1 Gag cores to make VLPs with human Her2 displayed on the surface. The glycosylation was also modified to include mammalian enzymes to generate Her2ma VLPs vs insect cell glycosylated Her2ic VLPs. Adjuvants PolyI:C or AddaVax were used.

Although the manuscript is thorough and the experimental methods are well defined I have major concerns regarding the manuscript

Major concerns:

The authors are using Human Her2 to immunize Balb/c mice and rat Her2 expressing TUBO cells for challenge. This model itself seems flawed due to the variability in the proteins from these species. We have to give credit to the authors for discussing it on line 562 and 563. Unfortunately this flaw makes the conclusions from these studies unreliable.

The authors should repeat the study using Human Her2 transgenic mice (Piechocki et al 2003).

The conclusion are not supported by the data. Similarly the title of the paper states that there is protection via the VLPs is also somewhat misleading as Human Her2 is used to immunize mice and the challenge is with Rat Her2 expressing carcinoma cells. Even with these limitations the protection is limited at best.

Minor comments:

Glycosylation pattern of Her2ma VLPs should be compared to VLPs generated in a human cell line.

Irradiated TUBO cells could be used as a control immunogen. This would clarify how immunogenic Rat Her2 would be in mice.

Addavax adjuvant should be described for the readers

It is not clear how the recombinant Her2 used in ELISA experiments was generated. Was it bacterial or made in insect cells. Would differences in the glycosylation pattern affect the binding?

The failure of Mammalian glycosylated VLPs generated by authors needs to be discussed further. Perhaps studies using viral antigens rather than Her2 would have provided better results.

Author Response

Response to Reviewer 3 Comments

Point 1: The authors are using Human Her2 to immunize Balb/c mice and rat Her2 expressing TUBO cells for challenge. This model itself seems flawed due to the variability in the proteins from these species. We have to give credit to the authors for discussing it on line 562 and 563. Unfortunately this flaw makes the conclusions from these studies unreliable

Response 1: There is 85% homology between the extracellular domains (ECD) and 88% between the full-length human and mouse HER2. Moreover, mouse and rat full-length HER2 show 95% homology and the ECD’s 94% homology. For better understanding and clarification we have investigated binding of serum antibodies to recombinant mouse HER2 and included ELISA data as supplementary material (Supplementary Figure 3).

We have included following sentence in the results section (line 476): “Furthermore, binding of serum antibodies to recombinant mouse HER2 was confirmed in an ELISA (Supplementary Figure 3), proofing the cross-reactivity of antibodies induced by VLP vaccination.

The following was added in line 577: “…and cross-reactivity of induced anti-HER2 antibodies was confirmed,…

Point 2: The authors should repeat the study using Human Her2 transgenic mice (Piechocki et al 2003).

Response 2: We agree, that our study can only provide a proof on principle, and based on our results we can now go a step forward and repeat the study with Human HER2 transgenic mice.

Point 3: The conclusion are not supported by the data. Similarly the title of the paper states that there is protection via the VLPs is also somewhat misleading as Human Her2 is used to immunize mice and the challenge is with Rat Her2 expressing carcinoma cells. Even with these limitations the protection is limited at best.

Response 3: Our main goal was to show the feasibility of VLPs to be able to be used as cancer vaccine, we believe putting more detailed information into the title does not contribute to the clarity of the manuscript and we would rather maintain the title as it is.

Point 4: Glycosylation pattern of Her2ma VLPs should be compared to VLPs generated in a human cell line

Response 4: In the present study our glycosylation analysis gives an estimate about the glycan structures of the tested proteins and indicates mammalianized glycosylation of HER2ma, we do not claim that it is identical with HER2 derived from CHO cells or HEK cell; there are differences in glycosylation in all different animal and mammalian expression systems; however we aimed to show whether any difference in glycosylation has an effect on immune response and protection. We agree that a comparison between the glycopatterns of HER2ma VLPs and VLPs generated in a human cell line would be a valuable contribution, however are beyond the scope of the presented work, but will be considered in follow-up studies.

Point 5: Irradiated TUBO cells could be used as a control immunogen. This would clarify how immunogenic Rat Her2 would be in mice

Response 5: We agree with the reviewer, that irradiated TUBO cells could have been used as control immnogen to clarify the immunogenicity of rat HER2 in BALB/c mice. Although, from literature we know that TUBO-based in vivo syngeneic tumor models have been extensively used in the field of anti-HER2 vaccine development, allowing HER2+ tumors to arise in the receptor animal without rejection or off-target effects (Boggio et al., 1996, PMCID: PMC2212479, Rovero et al. 2000, PMID:11046045).

According to literature “TUBO cells are a cloned line established in vitro from a BALB-neuT mouse mammary carcinoma. They display membrane class I H-2dMHC glycoproteins and rp185neu proteins. In BALB/c mice, r-p185 is a xenogeneic Ag that differs from mouse rp185 in <6% of the amino residues. Despite these differences, a challenge of 1 x 105 TUBO cells grew progressively in all BALB/c mice and gave rise to lobular carcinomas histologically similar to those that appear in BALB-neuT-transgenic mice. The reactive cell infiltrate associated with TUBO cell growth was marginal. No anti-TUBO cell CTL, nor IFN-gama, nor GM-CSF release were found when Spc from BALB/c mice bearing 3- or 10-mm mean TUBO tumors were tested immediately or after 6 days in in vitro restimulation with TUBO cells. Moreover, no anti-rp185 Ab were detected in the sera of tumor bearing mice. Despite their high membrane expression of the xenogeneic rp185, growing TUBO cells appear to trigger a marginal or no immune reaction in BALB/c mice.”

[Ref: http://www.ebi.ac.uk/ontology/webulous#OPPL_pattern ]

Point 6: Addavax adjuvant should be described for the readers

Response 6: AddaVax has been described for the readers.

Line 81 was rephrased: “…the immunogenicity and anti-tumor activity in conjunction with AddaVax (MF59-like squalene-based oil-in-water nano-emulsion)…”

Moreover, AddaVax has been described in the discussion section starting at line 611: “On the other hand, AddaVax is a squalene-based oil-in-water emulsion that elicits both cellular (Th1) and humoral (Th2) immune responses. AddaVax is believed to act through recruitment and activation of APCs and stimulation of cytokine production by macrophages and granulocytes [39].”

Point 7: It is not clear how the recombinant Her2 used in ELISA experiments was generated. Was it bacterial or made in insect cells. Would differences in the glycosylation pattern affect the binding?

Response 7: Recombinant human HER2 protein was expressed in High Five insect cells. For clarification line 245 in the Material and methods section was rephrased: “…HER2 protein (expressed in High Five insect cells) at a concentration of 2 µg/ml in phosphate...”

Also recombinant mouse HER2 was expressed in High Five insect cells and is described in the Figure legend of Supplementary Figure 3 (see Comment1/Response1).

Point 8: The failure of Mammalian glycosylated VLPs generated by authors needs to be discussed further. Perhaps studies using viral antigens rather than Her2 would have provided better results.

Response 8: In order to discuss the importance of human glycosylation of antigens, we rephrased following paragraph, starting at line 559, in the discussion section: ”It has been shown that the composition of glycans attached to therapeutic proteins potentially influences the efficacy and immunogenicity of the drug [32]. However, the impact of the glycosylation in case of HER2 studies remains poorly explored...”

The effects of different glycan structures is not predictable, but should be taken into consideration, especially if there are no previous studies.

For our study design, we believe that using viral antigens would not be optimal since viral antigens do not generate efficient immune response against HER2.

Round 2

Reviewer 1 Report

Response to (Response to Reviewer 1 Comments)

Comments are in italic.

Title: A HER2-displaying virus-like particle vaccineprotects from challenge with mammary carcinoma  cells in a mouse model 

Point 1: Figure 1: A SDS-PAGE for each of the three different VLPs preparation should be showed with the identification of each of the expected proteins. This is important to assess the level of contaminants in the vaccine preparation. It is an easy way to compare the content of the three samples.

Response 1: We have included a SDS-PAGE gel (Fig.1E), showing HIV-1 Gag and its degradation products and contaminants. Since merely the HIV-1 Gag in VLPs can be clearly seen in Coomassie staining and HER2 only in Western Blot analysis, we have additionally focused in more detail on the quantification of HER2 protein on the surface of VLPs in this work (indicated in Table 2).

Comment: The SDS-gel confirmed that the protein samples are highly contaminated with protein from the insect cells and the baculovirus vector used to express the VLPs. 

            The authors pretend that the HER2 antigen is found at the surface of the VLPs, but I do not see any evidence in this manuscript that support this statement. Based on the results presented, the authors can only conclude that the HER2 antigen is found with the same fraction of the VLPs.

Point 2:Figure 1: The quality of the western blot is not acceptable. A full gel should be showed for the western blots directed to HER2 and to the HIV Gag to assess for the degradation products and the overall quality of the sample that have been immunized.   

Response 2:We replaced Figure 1 E by a SDS-PAGE gel showing HIV-1 Gag (see above) and three full Western Blots showing HIV-1 Gag, HER2 and HER2_trunc.The Figure legend (line 366) was changed to: “(E) SDS-PAGE and Western blot analysis results of the VLP preparations are shown. Trastuzumab, a monoclonal anti-HER2 antibody (3B5) and monoclonal rabbit anti-human HIV-1 Gag antibody were used to probe truncated HER2, full-length human HER2 protein and HIV-1 Gag matrix protein, respectively.

Line 151 was rephrased: “…Aldrich, St. Louis, Missouri, USA). Detection of truncatedhuman HER2 was conducted using…”

A sentence was added in line 153: “Full-length HER2 was detected using anti-HER2 mAb (3B5; Thermo Fisher, MA5-13675) and anti-mouse IgG alkaline phosphatase-labeled antibody (Sigma, A3438).”

Line 351 was rephrased: ”…analysis using anti-human HER2 mAb (3B5),Trastuzumab and anti HIV-1 Gag rabbit antibody, as…”

Comment: Good.

Point 3: The cleavage product of HER2 is of different molecular weight between the HER2ic and the HER2ma. Is it real or is it a gel effect because one of the two samples was overloaded?

Response 3: Results indicate that glycosylation is responsible for the difference in size since glyco-analysis showed a defined difference between mammalianized and insect cell glycosylated preparations (indicated in Table 1 and Figure 2), and calculated molecular weight is in accordance with the experimental results. Moreover we have never before observed a shift of HER2 in molecular weight, due to overloading; here same amounts of VLPs have been loaded onto the gel. 

Comment: OK therefore, you should mention it in the result section.

Point 4: Figure 2: Several glycans (like Man 4 to 6) described at table 1 are not identified at Figure 2. Either the author should delete them from the table because they are not present in sufficient amount to be discussed or their position at Fig 2 should be highlighted like the others.

Also, there is no way to assess if the glycans issued from this digestion are derived from the VLP, the contaminant baculovirus or any other contaminating proteins in the vaccine preparation. Therefore, it is very difficult to draw any conclusion from this assessment

Response 4: Figure 2 was changed accordingly. All glycans present in Table 1 were annotated in Figure 2.

We agree that it is difficult to assess whether glycans are derived from the VLP, baculovirus or any other contaminating protein in the vaccine preparations, however for detecting alterations in glycan patterns towards mammalian like structures, all N-glycosylated proteins display such modifications, whether derived from VLP, baculovirus or any other contaminant. We can therefore assume that VLPs also display proteins with mammalianized modifications. Moreover, the sample subjected to MS analysis was exactly the same as was injected into mice. All identified glycans were present in the preparations used for vaccination.  

Comment: I agree with your response and I believe you should mention it in the result section. You should state that you assume that the glycosylation pattern is modified on the VLPs, even if you do not have the proof that it is the case. 

Point 5: Table 2: The curve of the DLS should be presented. It is anticipated that at least two peaks will be observed, on for the VLPs (about 100nm) and one for the contaminating baculovirus (about 260nm). The authors should demonstrate the homogeneity of the samples. The curves can be placed in Suppl. data

Response 5: We have included the curves of the DLS as supplementary material (Supplementary Figure 1); however, just a single peak is visible, due to the log scale on the x-axis, which is automatically generated by the Zetasizer softwear. In our study we wanted to demonstrate homogeneity of the sample, no cell debris or bigger aggregates are visible. The curves indicate a high quality biological sample, PDI is included in table 2 and is below 0,2 confirming homogeneity.

Line 394 was rephrased: “…VLPs were homogeneously distributed (PDI < 0,2; DLS diagrams see Supplementary Figure 1).”

Comment: A PDI of 0,2 indicates that the sample is not homogenous. The PDI should be inferior to 0,1 to be considered as homogenous. It should be changed for a sample that is moderate polydisperse which refers to a PDI between 0.1-0.4. See to Malvern  web site (https://www.materials-talks.com/blog/2017/10/23/polydispersity-what-does-it-mean-for-dls-and-chromatography/)

Point 6: Table 2: This table reveal that HER2ic contains at least 2 times more contaminant baculovirus than the HER2ma. As mentioned in the discussion, the baculovirus can influence the immunogenicity of the samples. To be comparable, the amount of contaminating baculovirus between the two samples should be standardized and made equivalent between the samples. Otherwise, it is impossible to assess the effect of the glycosylation pattern on the immunogenicity.

Response 6: We agree, that significant differences in BV concentration might influence the immune response, however based on our experience we strongly believe it’s highly unlikely that a 2-fold BV concentration is biologically significant. Studies in the past have shown (Margine et al., 2012, DOI:10.1371/journal.pone.0051559) that there is a difference whether BV are present or completely absent, but small differences in BV concentrations in vaccine samples had so far minor to no influence. 

Moreover, vaccine preparations were standardized based on total protein concentration, see Response 7.

Comment: To support your argument, you should then measure and compare, with comparable units, the amount of VLPs concentration vs the baculovirus concentration. As it is showed in table 2, the PFU and particles/mL can not compared and you cannot conclude that the amount of contaminant baculovirus is small. A quantitative ELISA for the VLPs and the baculovirus should be done to compare them with the same unit, like in µg/mL.

To my point of view this is really critical because it is well known that baculoviruses are very immunogenic. They possess an adjuvant property that is related to their capacity to elicit innate immunity in mice and induction of the secretion of interferon type 1 (Hervas-Stubbs, S., Rueda, P., Lopez, L., Leclerc, C., 2007. Insect baculoviruses strongly potentiate adaptive immune responses by inducing type I IFN. J. Immunol. 178, 2361e2369.) These a long list of authors that have already reported the adjuvant property of baculovirus.(Kitajima, M., Hamazaki, H., Miyano-Kurosaki, N., Takaku, H., 2006. Biochem. Biophys. Res. Comm. 343 (2), 378e384.Kitajima, M., Takaku, H., 2008. Clin. Vaccine Immunol. 15 (2), 376e378Abe, T., Takahashi, H., Hamazaki, H., Miyano-Kurosaki, N., Matsuura, Y., Takaku, H., 2003. J. Immunol. 171, 1133e1139. etc…..).In fact, because of their adjuvant property, several groups used baculovirus as a vaccine platform (see review: Appl Microbiol Biotechnol. 2017 May;101(10):4175-4184. )

Point 7: In addition, this table reveal that there is at least 2 times less HER2 protein associated with the VLP of HER2ma than HER2ic. It is likely that a 2-fold difference in the content of HER2 between Her2ma and HER2ic can have a major impact on the immune response that will be triggered. Therefore, a group of mice immunized with at least 2 times more HER2ma should have been included in the study to balance the amount of HER2 injected in the animals

Response 7: VLPs were standardized based on total protein content. Total protein content of HER2ma was lower than HER2ic (indicated in Table 2); therefore higher amounts were vaccinated, suggesting roughly the same content of HER2 in the vaccine preparations. According to ELISA of serum samples, we detected high anti-HER2 antibody titres in both, HER2ic and HER2ma vaccinated animals. Moreover, immune responses do not follow a linear pattern and small differences in HER2 are not expected to cause a significant difference in the immune response.

Comment: Based on table 2, the amount of HER2 per 100µg of total protein for HER2ic is 7,45(62.62/8.4) and is 4,7 (29,10/6.2) for HER2ma. This is a difference of 1,6 fold. To my point of view, this is a major difference in the amount of HER2 between the two samples. 

            It is impossible based on the data presented in this manuscript to discriminate HER2ic and HERma if the amount of antigen used for the immunization is not comparable. 

            This manuscript presents only preliminary data on the immunogenicity of these constructs. A dose response should be made to assess the immunogenicity of the two vaccine preparations before to make conclusions.

Point 8: Figure 3: The quality of the image is too low and it is difficult to see the results presented

Response 8: Figure 3 was replaced accordingly.

Comment: OK

Point 9: To highlight the benefice of the presentation of HER2 onto by the VLP, a control group with purified HER2 alone should be included. As it is presented here, it is not possible to judge if the VLP are improving the immune response directed to HER2

Response 9: We agree that it would be beneficial to show the improvement of the immune response when using VLP-displayed HER2 compared to purified HER2. However, in our study we only compared different VLP samples in conjunction with different adjuvant substances, and can’t judge whether our samples are more effective as compared to recombinant HER2. Yet, there are studies (Palladini et al., 2018,doi:10.1080/2162402X.2017.1408749) that have demonstrated that the immune response to VLPs was higher as compared to HER2 DNA vaccines.

Comment: When I read the title of the manuscript: ‘A HER2-displaying virus-like particle vaccineprotects from challenge with mammary carcinoma  cells in a mouse model ‘, I am somehow expecting the authors to demonstrate the advantage of the presentation of the HER2 antigen on the VLPs since the system was chosen for this purpose. 

Point 10: What is the proportion of the HER2 that is located at the surface of the VLP with the HER2ic and HER2ma constructs and how much of HER is free I in the vaccine preparation? This should be investigated and answered to conclude on the potential of the VLP in the improvement of the immune response to HER2

Response 10: We agree that knowing the proportions of VLP incorporated HER2 and free HER2 would be beneficial to conclude on the potential of the VLP in the improvement of the HER2 specific immune response. Although, after ultracentrifugation we do not expect a significant quantity of free HER2 to be present in the samples. To clarify, detailed characterization of the VLP samples in context with their immune effects will be subject to future studies.

Comment: To my point of view, this assessment is important for a better understanding of the results obtained in vivo.

Point 11: The adjuvant Addavax is a oil in water adjuvant (MF-59 like adjuvant). Since the VLPs are enveloped with the cell membrane and since the enveloped is necessary to present the HER2 protein at their surface, the impact of the adjuvant on the structure of the VLP should be investigated. Does it disrupt the membrane of the VLPs? Does it release the HER2 from the VLP? Does it chance the antigenic presentation of the HER2 antigen?

Response 11: Such detailed studies would provide a valuable contribution, however are beyond the scope of the presented work. Oil in water suspension is similar to lipid membranes which surround the VLPs and should not be destructive, yet even if their structure is destroyed, we would expect that still intact HER2 would be present in the samples. Still, the question is highly interesting and follow up studies will be done in order to investigate the impact of AddaVax on the structure of the VLP.

Comment: I disagree with the author. In addition, the impact of addavax on the structure of the VLPs is easy to perform. A sucrose density gradient in presence of the adjuvant supported by electron microscopy and/or DLS should rapidly give the answer. This data from such an experiment could change the interpretation of the results obtained in vivo.

Point 12: The authors chose to use only 5 mice per group for the immunization and the assessment of the anti-cancer efficacy of the vaccine. The experiment should at least include 10 mice per group and be repeated twice to provide a reasonable level of confidence

Response 12: The mouse experiments were exploratory, we agree that more mice would be more representative, however our sample was enough to be hypothesis generating in nature, follow-up studies with successful candidates will use a higher number of mice.

Comment: If the experiments in mice are exploratory, you should mention in in the manuscript and you should also explain that you only consider these results as a direction for future in cancer immunotherapy. To my personal point of view, exploratory results are not ready to be published in this journal. 

Point 13: There is 85% homology between the human and the mice HER2. Are the differences located in the surface exposed region of the protein? Are the antibody directed to the human HER2 capable to bind to the mouse HER2? This is critical to understand the mechanism of action of the vaccine candidates. This should be demonstrated

Response 13: There is 85% homology between the extracellular domains and 88% between the full-length human and mouse HER2. 

For better understanding and clarification we have investigated binding of serum antibodies to recombinant mouse HER2 and included ELISA data as supplementary material (Supplementary Figure 3). Mouse serum against human HER2 showed binding to mouse HER2, indicating that human anti HER2 antibodies recognize recombinant mouse HER2 and are cross-reactive. In contrast to monoclonal antibodies, immunized sera contain a large variety against the antigen.

We have included following sentence in the results section (line 476): “Furthermore, binding of serum antibodies to recombinant mouse HER2 was confirmed in an ELISA (Supplementary Figure 3), proofing the cross-reactivity of antibodies induced by VLP vaccination.”

The following was added in line 577:“…and cross-reactivity of induced anti-HER2 antibodies was confirmed,…

Comment: Good

Point 14: What is the contribution of the CTL response directed toward HER2 in the anti-cancer activity recorded? Could you repeat this experiment in mice where the CD8+ cells are depleted before implantation of the cancer cells?

Response 14: We have addressed this issue in the Discussion (see line 614):

“It has been previously shown that cytokines like IL-12 are required for the establishment of a long-term protection and tumor-specific memory against mammary adenocarcinomas which is mediated by CD8+ lymphocytes [33]. As we have previously noted, HER2ic-AddaVax immunized tumor biopsies have shown an increased presence of tumor infiltrating CD8+ lymphocytes that could contribute to the long-term response observed in this group.”

Experiments with CD8+ depleted mice would be very interesting, however cannot be accomplished within the scope of this work, but will be subject of further studies.

Comment: I disagree and I believe they should be included in this study.

Additionnal comment: Finally, the addition of vaccine preparation that has been exposed to a glycosidase should be included in the study before to conclude. It will help considerably to interpret the results. 

Author Response

Title: A HER2-displaying virus-like particle vaccine protects from challenge with mammary carcinoma cells in a mouse model 

Responses are in red.

Point 1: Figure 1: A SDS-PAGE for each of the three different VLPs preparation should be showed with the identification of each of the expected proteins. This is important to assess the level of contaminants in the vaccine preparation. It is an easy way to compare the content of the three samples.

Response 1: We have included a SDS-PAGE gel (Fig.1E), showing HIV-1 Gag and its degradation products and contaminants. Since merely the HIV-1 Gag in VLPs can be clearly seen in Coomassie staining and HER2 only in Western Blot analysis, we have additionally focused in more detail on the quantification of HER2 protein on the surface of VLPs in this work (indicated in Table 2).

Comment: The SDS-gel confirmed that the protein samples are highly contaminated with protein from the insect cells and the baculovirus vector used to express the VLPs. 

            The authors pretend that the HER2 antigen is found at the surface of the VLPs, but I do not see any evidence in this manuscript that support this statement. Based on the results presented, the authors can only conclude that the HER2 antigen is found with the same fraction of the VLPs.

Response: In Point 1 the reviewer requested that a SDS-PAGE should be included that identifies the expected protein, which we did. There was no request to analyse the HER2 on the surface of the VLPs, and we agree that a SDS-PAGE is not suitable for this kind of analysis. We have previously published a paper, where the quality and quantity of HER2 on the surface of VLPs was investigated in detail by TEM immuno-gold staining and biolayer interferometry-binding assays (Nika et al., doi:10.1016/j.pep.2017.06.005), showing that HER2 is present on the VLP surface, therefore, we do not pretend, instead, we have strong evidence that HER2 is present on the surface of VLPs.

Point 2:Figure 1: The quality of the western blot is not acceptable. A full gel should be showed for the western blots directed to HER2 and to the HIV Gag to assess for the degradation products and the overall quality of the sample that have been immunized.   

Response 2:We replaced Figure 1 E by a SDS-PAGE gel showing HIV-1 Gag (see above) and three full Western Blots showing HIV-1 Gag, HER2 and HER2_trunc.The Figure legend (line 366) was changed to: “(E) SDS-PAGE and Western blot analysis results of the VLP preparations are shown. Trastuzumab, a monoclonal anti-HER2 antibody (3B5) and monoclonal rabbit anti-human HIV-1 Gag antibody were used to probe truncated HER2, full-length human HER2 protein and HIV-1 Gag matrix protein, respectively.”

Line 151 was rephrased: “…Aldrich, St. Louis, Missouri, USA). Detection of truncatedhuman HER2 was conducted using…”

A sentence was added in line 153: “Full-length HER2 was detected using anti-HER2 mAb (3B5; Thermo Fisher, MA5-13675) and anti-mouse IgG alkaline phosphatase-labeled antibody (Sigma, A3438).”

Line 351 was rephrased: ”…analysis using anti-human HER2 mAb (3B5),Trastuzumab and anti HIV-1 Gag rabbit antibody, as…”

Comment: Good.

Point 3: The cleavage product of HER2 is of different molecular weight between the HER2ic and the HER2ma. Is it real or is it a gel effect because one of the two samples was overloaded?

Response 3: Results indicate that glycosylation is responsible for the difference in size since glyco-analysis showed a defined difference between mammalianized and insect cell glycosylated preparations (indicated in Table 1 and Figure 2), and calculated molecular weight is in accordance with the experimental results. Moreover we have never before observed a shift of HER2 in molecular weight, due to overloading; here same amounts of VLPs have been loaded onto the gel. 

Comment: OK therefore, you should mention it in the result section.

Response: It is mentioned in the results section, line 356: “HER2 with mammalianized glycosylation appeared heavier than the insect cell glycosylated counterparts, indicating the presence of additional glycans.”

Point 4: Figure 2: Several glycans (like Man 4 to 6) described at table 1 are not identified at Figure 2. Either the author should delete them from the table because they are not present in sufficient amount to be discussed or their position at Fig 2 should be highlighted like the others.

Also, there is no way to assess if the glycans issued from this digestion are derived from the VLP, the contaminant baculovirus or any other contaminating proteins in the vaccine preparation. Therefore, it is very difficult to draw any conclusion from this assessment

Response 4: Figure 2 was changed accordingly. All glycans present in Table 1 were annotated in Figure 2.

We agree that it is difficult to assess whether glycans are derived from the VLP, baculovirus or any other contaminating protein in the vaccine preparations, however for detecting alterations in glycan patterns towards mammalian like structures, all N-glycosylated proteins display such modifications, whether derived from VLP, baculovirus or any other contaminant. We can therefore assume that VLPs also display proteins with mammalianized modifications. Moreover, the sample subjected to MS analysis was exactly the same as was injected into mice. All identified glycans were present in the preparations used for vaccination.  

Comment: I agree with your response and I believe you should mention it in the result section. You should state that you assume that the glycosylation pattern is modified on the VLPs, even if you do not have the proof that it is the case. 

Response: We agree with the reviewer and included an additional paragraph in the results section, line 381:”Since a whole glyco pattern analysis was done it is difficult to assess whether glycans are derived from the VLPs, contaminating baculovirus or any other protein impurity in the vaccine preparations. Still, for detecting alterations in glycan patterns towards mammalian-like structures, all N-glycosylated proteins display desired modifications, suggesting that VLPs also display proteins with mammalianized alterations.”

Point 5: Table 2: The curve of the DLS should be presented. It is anticipated that at least two peaks will be observed, on for the VLPs (about 100nm) and one for the contaminating baculovirus (about 260nm). The authors should demonstrate the homogeneity of the samples. The curves can be placed in Suppl. data

Response 5: We have included the curves of the DLS as supplementary material (Supplementary Figure 1); however, just a single peak is visible, due to the log scale on the x-axis, which is automatically generated by the Zetasizer softwear. In our study we wanted to demonstrate homogeneity of the sample, no cell debris or bigger aggregates are visible. The curves indicate a high quality biological sample, PDI is included in table 2 and is below 0,2 confirming homogeneity.

Line 394 was rephrased: “…VLPs were homogeneously distributed (PDI < 0,2; DLS diagrams see Supplementary Figure 1).”

Comment: A PDI of 0,2 indicates that the sample is not homogenous. The PDI should be inferior to 0,1 to be considered as homogenous. It should be changed for a sample that is moderate polydisperse which refers to a PDI between 0.1-0.4. See to Malvern  web site (https://www.materials-talks.com/blog/2017/10/23/polydispersity-what-does-it-mean-for-dls-and-chromatography/).

Response: We agree, according to the guidelines (dimensioned for designed particles, e.g. latex particles) a PDI of 0.15 (as it is for our VLP preparations) indicates a moderate polydisperse distribution. However, according to literature (Danaei et al., doi:10.3390/pharmaceutics10020057), values of 0.2 and below are most commonly deemed acceptable in practice for polymer-based nanoparticle materials. Even in drug delivery applications using lipid-based carriers, such as liposome and nanoliposome formulations, a PDI of 0.3 and below is considered to be acceptable and indicates a homogenous population.

Therefore we consider our nanoparticle preparations to be homogenously distributed. Still, in accordance with the guidelines, we have changed the wording to moderate polydisperse (PDI 0.1 - 0.4) in the manuscript.

Line 400 was changed to: “All particles had a mean diameter size of 140 ± 10 nm, the calculated PDI indicated a moderate polydisperse nanoparticle distribution (PDI 0.1-0.4).”

Table 2 (line 408) was changed accordingly. PDI (< 0.2) was changed to “PDI (0.1-0.4)”.

Line 419 was changed to: “A PDI between 0.1–0.4 refers to a moderate polydisperse suspension.”

Point 6: Table 2: This table reveal that HER2ic contains at least 2 times more contaminant baculovirus than the HER2ma. As mentioned in the discussion, the baculovirus can influence the immunogenicity of the samples. To be comparable, the amount of contaminating baculovirus between the two samples should be standardized and made equivalent between the samples. Otherwise, it is impossible to assess the effect of the glycosylation pattern on the immunogenicity.

Response 6: We agree, that significant differences in BV concentration might influence the immune response, however based on our experience we strongly believe it’s highly unlikely that a 2-fold BV concentration is biologically significant. Studies in the past have shown (Margine et al., 2012, DOI:10.1371/journal.pone.0051559) that there is a difference whether BV are present or completely absent, but small differences in BV concentrations in vaccine samples had so far minor to no influence. 

Moreover, vaccine preparations were standardized based on total protein concentration, see Response 7.

Comment: To support your argument, you should then measure and compare, with comparable units, the amount of VLPs concentration vs the baculovirus concentration. As it is showed in table 2, the PFU and particles/mL can not compared and you cannot conclude that the amount of contaminant baculovirus is small. A quantitative ELISA for the VLPs and the baculovirus should be done to compare them with the same unit, like in µg/mL.

Response: We have standardized samples according to total protein concentration; detailed quantification and characterization of BEVS produced particles is one of the biggest challenges of this system, we agree on that. While the amount of contaminating BVs might not be considered as small, the differences within the samples can be considered as small (2 fold).

To my point of view this is really critical because it is well known that baculoviruses are very immunogenic. They possess an adjuvant property that is related to their capacity to elicit innate immunity in mice and induction of the secretion of interferon type 1 (Hervas-Stubbs, S., Rueda, P., Lopez, L., Leclerc, C., 2007. Insect baculoviruses strongly potentiate adaptive immune responses by inducing type I IFN. J. Immunol. 178, 2361e2369.) These a long list of authors that have already reported the adjuvant property of baculovirus.(Kitajima, M., Hamazaki, H., Miyano-Kurosaki, N., Takaku, H., 2006. Biochem. Biophys. Res. Comm. 343 (2), 378e384.Kitajima, M., Takaku, H., 2008. Clin. Vaccine Immunol. 15 (2), 376e378Abe, T., Takahashi, H., Hamazaki, H., Miyano-Kurosaki, N., Matsuura, Y., Takaku, H., 2003. J. Immunol. 171, 1133e1139. etc…..).In fact, because of their adjuvant property, several groups used baculovirus as a vaccine platform (see review: Appl Microbiol Biotechnol. 2017 May;101(10):4175-4184. )

Response: In order to determine the exact effect of each single component, a whole range of additional studies must be carried out and would not be answered with quantification by ELISA. This is not possible, although there exists a quantitative ELISA for VLPs based on measuring HIV-1 Gag protein and calculating back to the number of VLPs (exact number of molecules is known and always the same), this is not available for BVs, since the structure and number of exact components of the capsid is not known. Using antibodies against the surface is not possible, since VLPs and BVs are both budded particles with a similar surface membrane.

We are aware that the effects of BVs in a sample on the immune system is quite complex, as is the effect of recombinant protein expression on a cell. Production batches will always differ, especially if the product a complex multi-subunit particle. Also, additional expression of glycosyltransferases has an impact on the overall expression capacity, leading to variations in amount of HER2 produced and present on the surface of VLPs and also on the production of infectious BV particles. In terms of a production process, conditions must be chosen to realistically be able to produce a product that is biologically effective. Changing the glycosylation has a major impact on the entire cell system, which might cause differences in product yield and the ratio of VLPs to BVs. The goal of this experiment was to test the feasibility of producing VLPs with modified glycosylation and to investigate the impact within the mouse study.

Point 7: In addition, this table reveal that there is at least 2 times less HER2 protein associated with the VLP of HER2ma than HER2ic. It is likely that a 2-fold difference in the content of HER2 between Her2ma and HER2ic can have a major impact on the immune response that will be triggered. Therefore, a group of mice immunized with at least 2 times more HER2ma should have been included in the study to balance the amount of HER2 injected in the animals

Response 7: VLPs were standardized based on total protein content. Total protein content of HER2ma was lower than HER2ic (indicated in Table 2); therefore higher amounts were vaccinated, suggesting roughly the same content of HER2 in the vaccine preparations. According to ELISA of serum samples, we detected high anti-HER2 antibody titres in both, HER2ic and HER2ma vaccinated animals. Moreover, immune responses do not follow a linear pattern and small differences in HER2 are not expected to cause a significant difference in the immune response.

Comment: Based on table 2, the amount of HER2 per 100µg of total protein for HER2ic is 7,45(62.62/8.4) and is 4,7 (29,10/6.2) for HER2ma. This is a difference of 1,6 fold. To my point of view, this is a major difference in the amount of HER2 between the two samples. 

            It is impossible based on the data presented in this manuscript to discriminate HER2ic and HERma if the amount of antigen used for the immunization is not comparable. 

            This manuscript presents only preliminary data on the immunogenicity of these constructs. A dose response should be made to assess the immunogenicity of the two vaccine preparations before to make conclusions.

Response: As mentioned in the manuscript, line 416:”The total protein concentration was quantified by Quickstart Bradford Dye Reagent (Bio-Rad, 5000201) with bovine serum albumin standard curve. Total HER2 protein was determined by quantitative ELISA.”

Our methods were applied in order to analyze the samples in as much detail as possible. We believe that the difference as calculated by the reviewer is negligible, since the antibody response after boost were comparable and could be considered as high.

A dose response would require additional experiment, however was not requested in Point 7.

Point 8: Figure 3: The quality of the image is too low and it is difficult to see the results presented

Response 8: Figure 3 was replaced accordingly.

Comment: OK

Point 9: To highlight the benefice of the presentation of HER2 onto by the VLP, a control group with purified HER2 alone should be included. As it is presented here, it is not possible to judge if the VLP are improving the immune response directed to HER2

Response 9: We agree that it would be beneficial to show the improvement of the immune response when using VLP-displayed HER2 compared to purified HER2. However, in our study we only compared different VLP samples in conjunction with different adjuvant substances, and can’t judge whether our samples are more effective as compared to recombinant HER2. Yet, there are studies (Palladini et al., 2018,doi:10.1080/2162402X.2017.1408749) that have demonstrated that the immune response to VLPs was higher as compared to HER2 DNA vaccines.

Comment: When I read the title of the manuscript: ‘A HER2-displaying virus-like particle vaccineprotects from challenge with mammary carcinoma  cells in a mouse model ‘, I am somehow expecting the authors to demonstrate the advantage of the presentation of the HER2 antigen on the VLPs since the system was chosen for this purpose. 

Response: In our opinion the title does not imply that HER2 displayed on the VLP surface is an advantage over soluble HER2, it just states that protection based on HER2-displaying VLPs was shown.

Point 10: What is the proportion of the HER2 that is located at the surface of the VLP with the HER2ic and HER2ma constructs and how much of HER is free I in the vaccine preparation? This should be investigated and answered to conclude on the potential of the VLP in the improvement of the immune response to HER2

Response 10: We agree that knowing the proportions of VLP incorporated HER2 and free HER2 would be beneficial to conclude on the potential of the VLP in the improvement of the HER2 specific immune response. Although, after ultracentrifugation we do not expect a significant quantity of free HER2 to be present in the samples. To clarify, detailed characterization of the VLP samples in context with their immune effects will be subject to future studies.

Comment: To my point of view, this assessment is important for a better understanding of the results obtained in vivo.

Response: We expect the amount of free HER2 to be not relevant, as after ultra-centrifugation, there is no free HER2 to be found, the only free HER2 could be derive from degradation, if the samples are old. In our study only freshly prepared samples were used. TEM pictures after sucrosegradient-ultracentrifugation show only intact VLPs, also we have previously shown (Nika et al., doi:10.1016/j.pep.2017.06.005) that Immunogold-labeling specifically stained HER2 protein on the surface of VLPs, no free HER2 was detected.

Point 11: The adjuvant Addavax is an oil in water adjuvant (MF-59 like adjuvant). Since the VLPs are enveloped with the cell membrane and since the enveloped is necessary to present the HER2 protein at their surface, the impact of the adjuvant on the structure of the VLP should be investigated. Does it disrupt the membrane of the VLPs? Does it release the HER2 from the VLP? Does it chance the antigenic presentation of the HER2 antigen?

Response 11: Such detailed studies would provide a valuable contribution, however are beyond the scope of the presented work. Oil in water suspension is similar to lipid membranes which surround the VLPs and should not be destructive, yet even if their structure is destroyed, we would expect that still intact HER2 would be present in the samples. Still, the question is highly interesting and follow up studies will be done in order to investigate the impact of AddaVax on the structure of the VLP.

Comment: I disagree with the author. In addition, the impact of addavax on the structure of the VLPs is easy to perform. A sucrose density gradient in presence of the adjuvant supported by electron microscopy and/or DLS should rapidly give the answer. This data from such an experiment could change the interpretation of the results obtained in vivo.

Response: Testing the impact of different adjuvants on our samples is not easy to perform and can only be done in a different study and only makes sense after proof of principle that shows that HER2 on VLPs has the principle capacity to induce protection, which we wanted to demonstrate first. There are many other interesting adjuvants that then might be tested, as is probably done for any vaccine candidate. Yet, this experiment is beyond the scope of this work and in our opinion would not contribute significantly to our results.

In our manuscript we wanted to call attention on the importance of assessing a suitable adjuvants for individual studies, which was mentioned in the manuscript, line 655: “We further demonstrated the importance of assessing the best adjuvant and showed that cell type specific glycan structures have an impact on the efficacy of a potential vaccine.”

Point 12: The authors chose to use only 5 mice per group for the immunization and the assessment of the anti-cancer efficacy of the vaccine. The experiment should at least include 10 mice per group and be repeated twice to provide a reasonable level of confidence

Response 12: The mouse experiments were exploratory, we agree that more mice would be more representative, however our sample was enough to be hypothesis generating in nature, follow-up studies with successful candidates will use a higher number of mice.

Comment: If the experiments in mice are exploratory, you should mention in in the manuscript and you should also explain that you only consider these results as a direction for future in cancer immunotherapy. To my personal point of view, exploratory results are not ready to be published in this journal. 

Response: As we already stated, our in vivo tumor challenge of pre-vaccinated mice were exploratory but hypothesis generating, allowing us to uncover important information for our future research approach: HER2ic-VLPs will be the candidate for further testing, AddaVax will be used as adjuvant.

For clarification we included a sentence in the conclusion section, line 658: “Thus, we consider these results as a direction for future cancer immunotherapy.”

Point 13: There is 85% homology between the human and the mice HER2. Are the differences located in the surface exposed region of the protein? Are the antibody directed to the human HER2 capable to bind to the mouse HER2? This is critical to understand the mechanism of action of the vaccine candidates. This should be demonstrated

Response 13: There is 85% homology between the extracellular domains and 88% between the full-length human and mouse HER2. 

For better understanding and clarification we have investigated binding of serum antibodies to recombinant mouse HER2 and included ELISA data as supplementary material (Supplementary Figure 3). Mouse serum against human HER2 showed binding to mouse HER2, indicating that human anti HER2 antibodies recognize recombinant mouse HER2 and are cross-reactive. In contrast to monoclonal antibodies, immunized sera contain a large variety against the antigen.

We have included following sentence in the results section (line 476): “Furthermore, binding of serum antibodies to recombinant mouse HER2 was confirmed in an ELISA (Supplementary Figure 3), proofing the cross-reactivity of antibodies induced by VLP vaccination.”

The following was added in line 577:“…and cross-reactivity of induced anti-HER2 antibodies was confirmed,…

Comment: Good

Point 14: What is the contribution of the CTL response directed toward HER2 in the anti-cancer activity recorded? Could you repeat this experiment in mice where the CD8+ cells are depleted before implantation of the cancer cells?

Response 14: We have addressed this issue in the Discussion (see line 614):

“It has been previously shown that cytokines like IL-12 are required for the establishment of a long-term protection and tumor-specific memory against mammary adenocarcinomas which is mediated by CD8+ lymphocytes [33]. As we have previously noted, HER2ic-AddaVax immunized tumor biopsies have shown an increased presence of tumor infiltrating CD8+ lymphocytes that could contribute to the long-term response observed in this group.”

Experiments with CD8+ depleted mice would be very interesting, however cannot be accomplished within the scope of this work, but will be subject of further studies.

Comment: I disagree and I believe they should be included in this study.

Response: Although mechanistically interesting, the results obtained in the future with CD8+ depleted mice will not change the actual observation that HER2ic-Addavax provided control of tumor growth and extended survival in pre-vaccinated mice. Narrowing the contribution of CTLs, and other immune cells with effector functions, to the already probed potential of HER2ic-Addavax is itself a new line of investigation that we will pursue in the future.

Additionnal comment: Finally, the addition of vaccine preparation that has been exposed to a glycosidase should be included in the study before to conclude. It will help considerably to interpret the results. 

Response: This experiment was not requested previously, and we do not see what we would learn from such an experiment, since HER2 is always glycosylated. Such an experiment would not require the production of VLPs.

Reviewer 3 Report

The authors have addressed the concerns I had.

Author Response

We thank the reviewer for critically evaluating our manuscript and accepting our revised version.